# CANDIDATE: A tool for generating anonymous participant-linking IDs in multi-session studies

**Frode Eika Sandnes**[1,2]*

**1** Department of Computer Science, Faculty of Technology, Art and Design, Oslo Metropolitan University, Oslo, Norway, **2** School of Economics and Information Technology, Kristiania University College, Oslo, Norway

* frodes@oslomet.no

## Abstract

### Background

To ensure the privacy of participants is an ethical and legal obligation for researchers. Yet, achieving anonymity can be technically difficult. When observing participants over time one needs mechanisms to link the data from the different sessions. Also, it is often necessary to expand the sample of participants during a project.

### Objectives

To help researchers simplify the administration of such studies the CANDIDATE tool is proposed. This tool allows simple, unique, and anonymous participant IDs to be generated on the fly.

### Method

Simulations were used to validate the uniqueness of the IDs as well as their anonymity.

### Results

The tool can successfully generate IDs with a low collision rate while maintaining high anonymity. A practical compromise between integrity and anonymity was achieved when the ID space is about ten times the number of participants.

### Implications

The tool holds potential for making it easier to collect more comprehensive empirical evidence over time that in turn will provide a more solid basis for drawing reliable conclusions based on research data. An open-source implementation of the tool that runs locally in a web-browser is made available.

**Data Availability Statement:** https://github.com/frode-sandnes/CANDIDATE.

**Funding:** The author received no specific funding for this work.

**Competing interests:** The authors have declared that no competing interests exist.

# Introduction

The privacy and safety of participants is of utmost importance in research that involves people. The World Medical Association's *Declaration of Helsinki–Ethical Principles for Medical Research Involving Human Subjects* states that "Every precaution must be taken to protect the privacy of research subjects and the confidentiality of their personal information." Privacy is also regulated by legislation such as the General Data Protection Regulations (GDPR) which applies in the European Union. When researchers need to store personal information such as names, national ID numbers, phone numbers, etc., the researchers need to document adequate mechanisms for secure storage of the data, and routines for deleting the data at the end of a project. Often researchers must file formal applications for permissions to store personal data. If personal data includes sensitive information, such as information about health, reduced functioning, and similar, the regulations and procedures are even stricter. Obviously, health related research often involves sensitive information. But studies in other fields such as computer science could also involve vulnerable cohorts such as individuals with dyslexia [1], low vision [2, 3] or other disabilities. The issue of privacy of participants in research studies is indeed highly relevant in many disciplines.

If a study can be conducted in a single session per participant, it is usually straightforward to maintain anonymity as one does not need to know the identity of the participant. For example, if a within-groups experiment where one observes how participants respond to two variations of an input technique can be conducted in a single session there is no need to know who these participants are [4]. Such anonymous data usually do not have privacy issues. The need to identify participants arises if one needs to consult participants several times. For example, if a within-groups experiment takes too long to be completed in a single session, one may have to split the various conditions of the experiment into several separately scheduled sessions. In a pre-post experimental design, one may have a first session where participants are probed following some intervention such as using some technology, and then a second session, scheduled later, where the participants are probed again to observe effects of the intervention [5]. Alternatively, to observe how participants learn to use a technology over time, one needs to observe the participants at various points in time at regular intervals [6, 7]. Longitudinal studies are used in several disciplines including physical health [8, 9], mental health [10], and human computer interaction [11–13]. The challenges associated with data linking for research has received much attention and it has been pointed out that erroneous record linkage may result in biased results [14]. Most of the research into record linkage has been of a technical nature. However, there have also been efforts to explore participants' perceptions of volunteering to participate in studies that require linking of data [15]. Findings showed that not all participants trust the anonymization mechanisms. Decisions to participate often rests on a balance between the sensitivity of the issues to be studied and the potential benefits the results may provide. When analysing multi-session data, it is necessary to link the data of one person in one session to the corresponding data in other sessions, to perform paired tests, repeated measures analyses, or similar analyses. Connecting data is straightforward with a linking table where each participant is assigned a running number. By labelling the data with the running number one avoid revealing the identity of that person in the data. One must assume that the linking table is kept confidential. However, if the linking table is leaked, the privacy of participants is compromised. The goal is therefore to avoid such linking tables altogether.

Linking tables are still commonly used but often require the researchers to solicit formal approvals. Acquiring such formal permissions can be time consuming, bureaucratic, and difficult requiring application writing experience and knowledge. Students who want to conduct a multi-session experiment during a course running over a semester may be prevented to do so

because of a lack of time and competence to obtain the necessary permissions. Students therefore miss out on the valuable learning experiences of conducting such experiments in practice. Moreover, it is a matter of concern if a researcher settles for a single session experiment when a multi-session experiment would be a more suitable choice simply to prevent the administrative and bureaucratic burden of obtaining formal permissions to store linking tables. Even worse, a researcher may choose to ignore the regulations and store personal information. A key motivation of this work thus is to simplify, or even eliminate the need for, formal data storage approval procedures.

Many anonymous linking procedures have been proposed during the last decades. They broadly fall into two categories, namely, self-generated codes and Bloom-filter approaches. Self-generated codes [16] rely on brief questionnaires the participants complete during each session. The responses to a set of personal questions are used to construct unique IDs for each participant. One drawback of self-generated codes is that they require effort from the participants diverting valuable attention away from the actual session activity. Moreover, participants may find the questions invasive. Self-generated codes have also been found to exhibit high error rates.

In contrast, Bloom filter approaches [17] are automatic and therefore used in large scale studies, especially studies involving register data. Typically, the bigrams making up the participants name are fed into a series of hash functions that are used to construct a bit vector. The use of bigrams means that the method is robust to input errors such as misspellings. One drawback of Bloom filter approaches is the very long and cryptic IDs, typically 1000 bits (or around 256 printable characters). Bloom filters have also been criticised for being vulnerable to attack [18].

Recent approaches attempted to generate short IDs that are perceivable less threatening to participants. For instance, the HIDE procedure [19] uses truncated hashes and a stochastic search for a universal salt (an encoding parameter) that results in a set of short and unique IDs. A salt is a piece of data (string) that is concatenated to a value before it is hashed. However, all the participants needed to be known in advance. To overcome this limitation the BRIDGE procedure was proposed [20] which also uses truncated hashes. But instead of searching for a salt that results in unique codes for a fixed set of names, the procedure detects collisions and asks the researcher to manually resolve collisions using word challenges. I.e., the researcher or participant must recognize, or not recognize, a given word.

Many of the published works on linking codes use the term "anonymous". Clearly, such schemes may be anonymous to an arbitrary onlooker. This understanding of "anonymous" will also be used herein. However, Oxford dictionary defines anonymity as a "lack of outstanding, individual, or unusual features". Using such a definition none of the published linking procedures can be considered fully anonymous as there are enough features stored such that someone, usually the participant, would be able to identify their own data (using their name or a self-generated code, etc). The "right to be deleted" GDPR principle can be used to illustrate this; If it is somewhat possible to satisfy a participant's request to have their data deleted, the data is not truly anonymous. Although linking procedures may not be able to provide anonymity in a strict sense, they may assist experimenters in designing robust data handling plans that reduce the risk of accidentally leaking personal information.

The goal of the CANDIDATE tool proposed herein was to overcome challenges with previous linking procedures by providing privacy through short IDs with an automatic procedure that does not rely on effort from the researcher and participants. The main emphasis in this study is to rely on participant's names as the basis for generating linking IDs. Soliciting additional information of a more private nature (such as the birth date) requires effort and time and may be perceived as intrusive and lead participants to withdraw from a study.

This paper is organized as follows. The next section reviews related work. Next, the procedure used by the CANDIDATE tool is presented followed by an evaluation and discussion of its integrity and anonymity. Finally concluding remarks are presented.

## Related work

Self-generated codes belong to one major anonymization category, and many such procedures have been proposed [21–27]. A self-generated code is created from the responses to a simple questionnaire the participants complete during each session. For example, a six-character ID could be generated from the first letter of the mother's name, number of older brothers, month of birth and the first letter of the middle name [28]. A study with 745 participants revealed that self-generated codes were successful in linking 75.2% of the records, 22.1% remained unmatched and 2.7% were matched incorrectly. Successful linking of just 3 of 4 participants may render this approach too unreliable. Similar results have been found in other studies [29] which concluded that self-generated codes are ineffective for longitudinal studies.

The alternative to self-generated codes are automatically generated codes. Several simple approaches have been proposed including attempts relying on the Soundex algorithm [30, 31]. The Soundex algorithm converts words (names) into a more coarse-grained phonetic representation. In this sense the Soundex algorithm is a one-way (irreversible) function. However, in practice Soundex does not provide anonymity as it is possible to match names according to their Soundex representations. It has been suggested that the anonymity with the Soundex encoding can be increased by obfuscating the IDs with additional dummy records [32]. The Soundex algorithm makes the procedure tolerant to input errors such as certain spelling mistakes [33–35].

Another simple approach is to encode participants using control numbers [36]. Such control numbers could be generated from pieces of demographic information such as first letter of surname, date of birth, gender, sum of the ASCII characters of the name, etc. The availability of demographic information may be limited in some contexts. More importantly, one should carefully question the actual anonymity of such schemes.

To improve the anonymity of participants several approaches rely on hash codes, either alone or in conjunction with other methods [37–39]. A hash function is a one-way function that provides few clues about the input, and it is therefore usually not possible to derive the input to the hash function based on the output (digest) of the hash function. However, simple hashing schemes are vulnerable to phonebook attacks [38] where one can confirm that a participant on a list of contenders was part of a study based on matching hash values. One way to overcome phonebook attack is to add a secret salt (a simple text string) to the name before hashing [40]. However, if this salt is leaked to adversaries the identities become vulnerable to phonebook attacks. Another approach is to reduce the information content such as just taking the two first characters from the first name and family names and date of birth as input to the hash function [41]. The consequence is that a phonebook attack will lead to multiple hits per ID and the adversary can therefore not conclude with certainty that a particular ID belongs to a particular person. A problem with this simple approach is that there is a probability of collision also for the names in the list of participants. The principle of obfuscating data such that multiple individuals share the same characteristics is known as $k$-anonymity [42].

During the last decade, the most intense research effort seems to have been in methods that rely on Bloom filters. Bloom filter approaches facilitate partial matches and are therefore robust to errors in the data [17]. However, critical voices have raised concerns over the anonymity of Bloom filters as they have been demonstrated to be vulnerable to cryptanalysis attacks [43–45]. Possible countermeasures include the use of salts with the hash functions [46].

Besides implementations provided by The German Record Linking Centre (https://www.record-linkage.de/) there appear to be few Bloom filter record linking implementations available to researchers. Bloom filters also exhibit long IDs. Such long IDs may appear cryptic to participants, and IDs comprising several hundred characters may be hard to work with during manual data handling as humans are typically able to store 5 items/digits at a time in short-term memory [47].

## The CANDIDATE anonymization tool

### Participant representations

It is assumed that the input to the procedure is the participant's full name (given name, middle name, and family name) transcribed in characters from the Latin alphabet, and that each name is unique. In most practical situations with small sample sizes this is usually the case, although an inspection of any phonebook will reveal that many individuals share the same names. Note that the procedure does not depend on the input being names but could comprise digits or combinations of digits and Latin characters. For example, national ID numbers could be used, but it is assumed that national ID numbers are perceived as more private and sensitive than names and participants may feel uncomfortable sharing their national ID numbers with researchers. Less sensitive alternatives could be the participants phone numbers or e-mail addresses. A researcher needs to assess what feature to use as the source for the ID encodings that is most beneficial to a given study. Note that the input error tolerance mechanisms do not apply If other representations than names are used. The current implementation of the tool does not support other character types such as Chinese, Russian, or Arabic.

Algorithm 1 shows the pseudo code of the CANDIDATE procedure and Fig 1A–1D shows corresponding flowcharts of the procedure. New participants are added with the *Add* procedure that takes the name of the participant, the anticipated maximum number of participants *L*, and the size of the coding space *N* as inputs. The size of the coding space needs to at least match the number of anticipated participants. The list of anonymous participant IDs are maintained in an ID-list. *Lookup* is used to find the ID of a given participant. *Encode* is used by *Add* and *Lookup* to transform names into IDs using one of the available hash functions.

```
Constants:

  hash-type_{default}: = 0

  hash-type_{offset}: = 10

  salt: = [salt_1, salt_2, ..., salt_n], n: = number of salts

  L: = maximum number of anticipated participants

  coding-factor: = 10 (typically)

State variable:

  ID-list: = {}

Add(name, L)

N: = coding-factor × L

  ID_{original}: = Encode(name, N, hash-type_{default})

  IF ID_{original} in ID-list
```

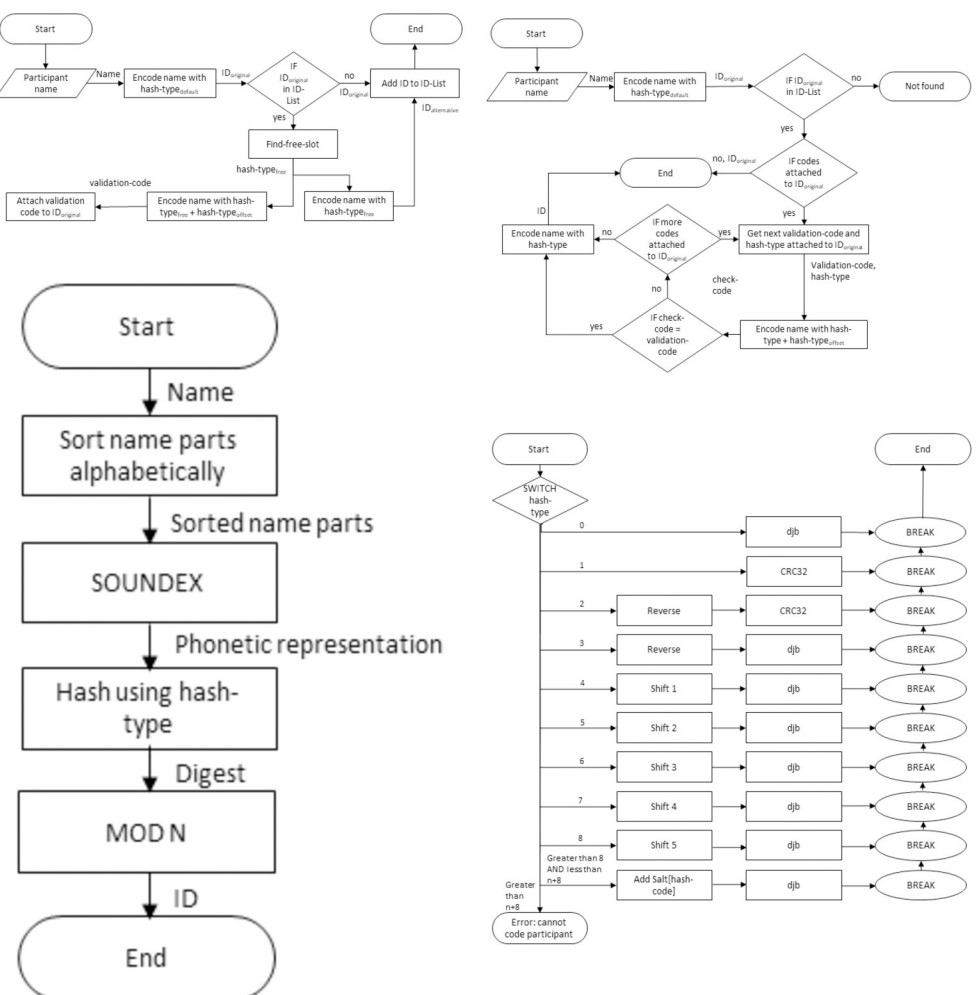

**Fig 1.** Flowcharts of the CANDIDATE procedure: (a) Add, (b) Encode, (c) Lookup, and (d) Hash.

```
hash-type_free: = Find-free-slot(name, L, ID-List)

ID_alternative: = Encode(name, N, hash-type_free)

validation-code: = Encode(name, N, hash-type_free + hash-type_offset)

ATTACH (hash-type_free, validation-code) TO ID_original

ID_original: = ID_alternative

ADD ID_original TO ID-list

Lookup(name, L)

N: = coding-factor × L

ID_original: = Encode(name, N, hash-type_default)

ID: = D_original

FOR EACH (hash-type, validation-code) ATTACHED TO ID_original
```

```
        ID_contender: = Encode(name, N, hash-type)

        ID-contender-validation: = Encode(name, N, hash-type +
    hash-type_offset)

        IF (ID-contender-validation == validation-code)

          ID: = ID_contender

      RETURN ID

  Encode(name, N, hash-type)

    name_sorded: = Sort(name)

    name_phonetic: = Soundex(name_sorted)

    digest: = Hash(name_phonetic, hash-type)

    ID: = digest MOD N

    RETURN ID

  Hash(name, hash-type)

    SWITCH (hash-type)

      0: digest: = djb2(name)

      1: digest: = CRC-32(name)

      2: digest: = CRC-32(reverse(name))

      3: digest: = djb2(reverse(name))

      4: digest: = djb2(shift(name, 1)

      5: digest: = djb2(shift(name, 2)

      6: digest: = djb2(shift(name, 3)

      7: digest: = djb2(shift(name, 4)

      8: digest: = djb2(shift(name, 5)

      ≥ 9 AND < n + 8: digest: = djb2(name + salt[8—hash-type])

      ≥ n + 8: UNRECOVERABLE ERROR—Cannot compute hash.

    RETURN digest
```

**Algorithm 1. Pseudo-code of the CANDIDATE linking algorithm (implementation available at https://github.com/frode-sandnes/CANDIDATE).**

## Encoding participants

*Add* attaches new participants and resolves collisions. The algorithm assumes that the input contains no unwanted characters such as hyphens and apostrophes. Such characters should be eliminated in the user interface. First, two optional steps can be applied if a name representation is used, namely sorting and phonetic coding. Name sorting involves sorting name parts (given, middle and family names) into alphabetical order to make the procedure robust to variations in name orderings. Next, the sorted name parts are converted into a phonetic

representation using Soundex. For example, "Christian" would be coded as C6235, i.e., the first letter (C), 6 for the r-sound, 2 for the c/g,/j,/k/q/s/x/z-sounds, 3 for the t/d-sounds and 5 for the m/n-sounds. Note that the full-length encoding is used, which differs from the original Soundex algorithm which only includes the first four characters (C623). This step makes CANDIDATE tolerant to certain input errors such as spelling mistakes or transcription errors. In short Soundex removes all vowels and double consonants, reassigns consonants to a more coarse-grained set of classes of similar sounding sounds, i.e., *m* and *n*, *d* and *t*, etc. Note that the Soundex step should only be used with name representations transcribed in Latin characters. Since Soundex is well-documented (see for instance [19, 30–35]), with many available implementations, it is not described in detail herein. Note that the phonetic coding and sanitation step can be switched off in the CANDIDATE implementation.

The phonetic representation is then hashed using one of several hashing functions. *Hash* returns the corresponding digest for a name using one of the available hash functions constructed from the djb2 and CRC-32 algorithms, namely a djb2-hash, CRC-32 code, CRC-32 of the name reversed, djb2 hash of the name reversed, djb2 hash of the characters of the names shifted by 1–5 characters, and djb2 hash of the names with salts taken from the most frequent words in English. Dan J. Bernstein's djb2 algorithm [48] builds a hash by processing the input from left to right. A hash is accumulated by adding the ASCII value of each character to the previous result multiplied by 33. CRC-32 (Cyclic redundancy check) [49] is more complex. It involves performing repeated XOR operations over a table of 256 32-bit precomputed constants for each character of the input. Fig 2 shows examples of how the string "Christian" is coded with the different hash functions.

This step ensures anonymity in that it should not be possible to identify who has participated in a study. The outputs of any linking procedure are linking IDs. These linking IDs must be assumed as being public and observable by adversaries. A procedure therefore needs to be designed such that it is impossible to derive the identity of a participant from the linking ID. In the CANDIDATE procedure anonymity of participants is ensured by applying a hash function to the name and truncating the digest (the output from the hash function). The truncation of the digest obfuscates the identity of a person as several different individuals will result in the same ID, and an attacker can therefore not be certain who a particular person is through phonebook attacks. The more the digest is truncated the higher the degree of anonymity is

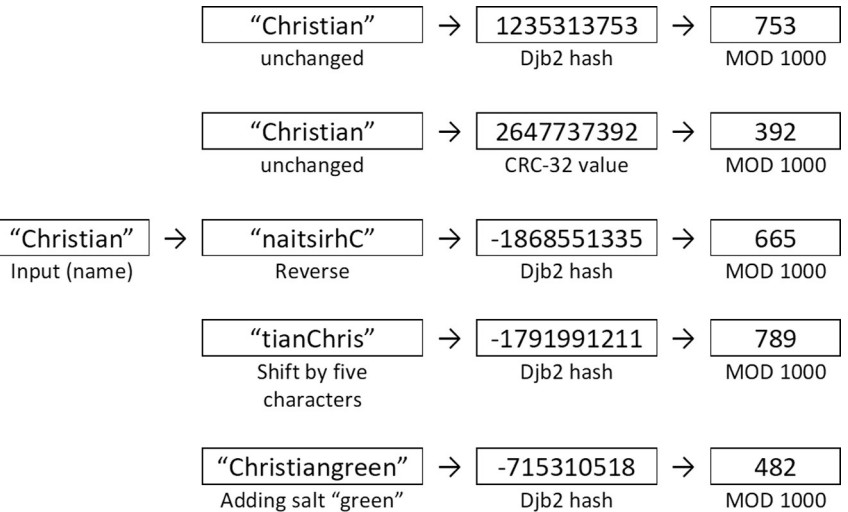

**Fig 2. Examples of how different hash functions code the name "Christian".**

achieved. CANDIDATE simply truncates by taking modulus $N$ of the digest (the reminder of the division of the hash output divided by $N$).

## Handling collisions

Each participant should be associated with a unique ID (*integrity*). However, there is a probability that a hash function will result in collisions. Table 1 illustrates the probability of collision with three different hash functions when coding 100 different names (based on a simulation with 10,000 iterations). Clearly, the probability of collision is related to the size of the coding space as larger coding spaces yield lower probabilities of collision. If the coding space has the same size as the number of items, there will be a collision for more than every third item.

Moreover, the larger the truncated part is the higher the probability of collision will be. There is thus a trade-off between the degree of anonymity and probability of collision. Collisions between items are desirable if the names are not part of the study as it strengthens anonymity, while we need to be able to uniquely distinguish between all participants in the sample. The CANDIDATE collision handling is based on two assumptions, namely that a) the list of existing IDs is known, and b) the researchers know that they are adding a new participant or looking up an existing participant. When adding a new participant, we first compute $ID_{original}$ using the default hash function and check the list of existing IDs to see if $ID_{original}$ is already used. If the ID does not exist, we have no collision, and the ID can be added and used as is. If the $ID_{original}$ already exists, we need to calculate an alternative ID, termed $ID_{alternative}$. This is done by applying a different hash function. We then need to check that the alternative ID is not occupied as well. It may therefore be necessary to try several different hash functions hash-type$_{free}$ in order to find an unused ID. For this purpose, *Find-free-slot* searches through an array of hash functions (djb2, CRC-32, djb2-reverse, . . .) that results in a unique match. Once we have found a hash function hash-type$_{free}$ that results in an unused ID, we associate $ID_{original}$ with the hash function hash-type$_{free}$ used to obtain the $ID_{alternative}$. $ID_{alternative}$ is added to the list of IDs and used for linking the participant. In addition, another hash function hash-type$_{free}$ + hash-type$_{offset}$ is used to calculate a validation code for the given name, and this validation code is also associated with $ID_{original}$. As multiple collisions may occur for an ID the pairs of hash functions and validation codes are connected. The hash function used to calculate the validation code must be different to the hash function used to calculate $ID_{original}$ and $ID_{alternative}$. The example in Fig 3 illustrates how collisions are handled using multiple hash functions.

**Table 1. Collision probability with djb2, CRC-32 and double hashes (half djb2/half CRC-32) for 100 randomly selected names with different coding space sizes.**
Based on simulation with 10,000 iterations.

| coding space | djb2 | CRC-32 | djb2+CRC-32 |
|---|---|---|---|
| 100 | 37.2238% | 37.2224% | 37.2198% |
| 1,000 | 4.8574% | 4.8584% | 4.8911% |
| 10,000 | 0.4966% | 0.5086% | 0.5084% |
| 100,000 | 0.0531% | 0.0496% | 0.0597% |
| 1,000,000 | 0.0042% | 0.0056% | 0.0064% |
| 10,000,000 | 0.0002% | 0.0004% | 0.0004% |
| 100,000,000 | 0.0000% | 0.0000% | 0.0000% |

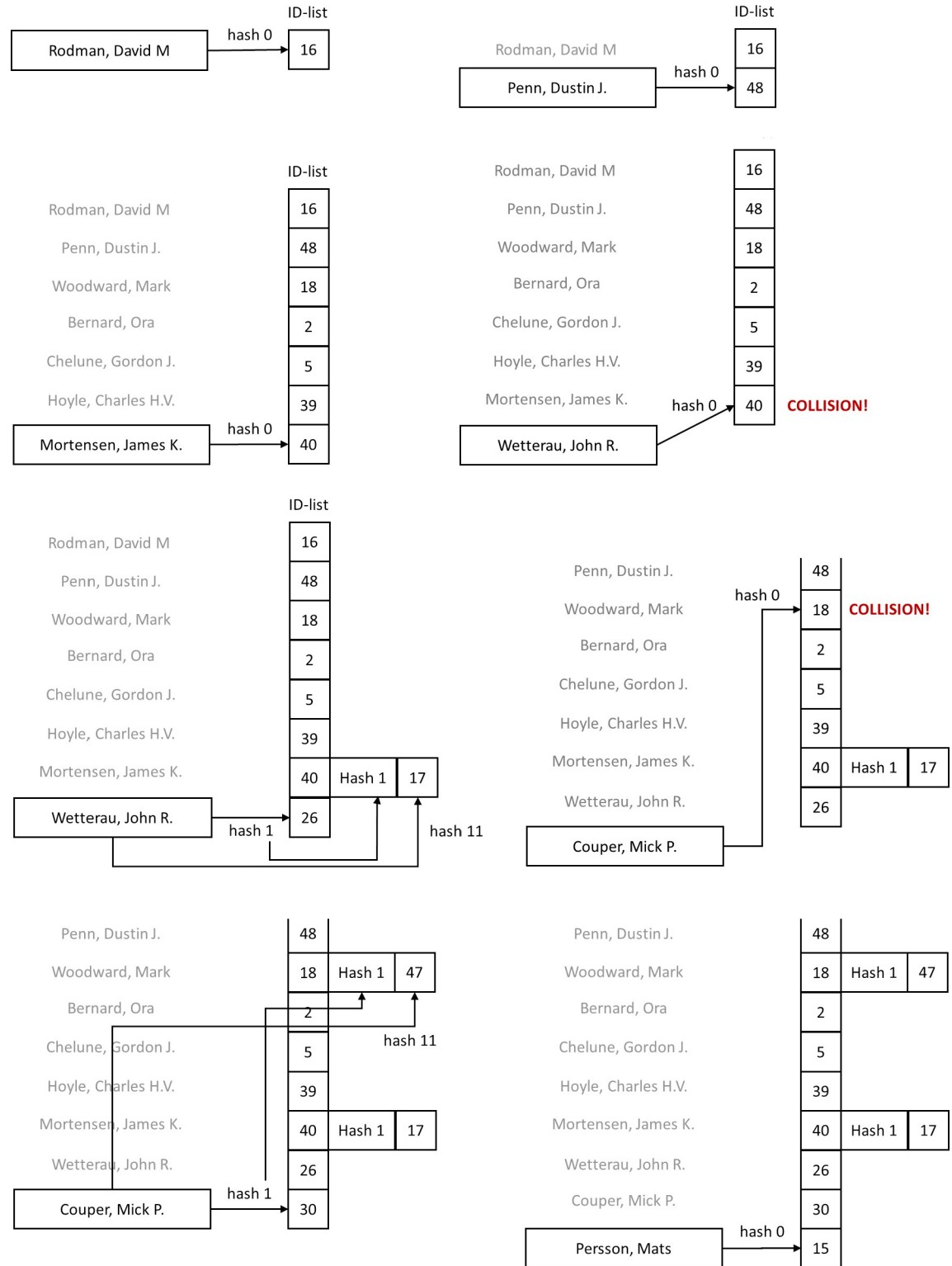

**Fig 3.** Example of encoding and collision handling: (a) Encoding the first participant with hash function 0. (b) Encoding the second participant with hash function 0. (c) Further seven participants encoded without any collisions. (d) The eighth participant gives a collision. (e) A free slot is found by instead using hash function 1 (CRC-32). The hash function used (hash function 1) is attached to the collision entry together with a check code obtained using hash function 11 (salted). (f) A collision also occurs for participant nine. (g) A free slot is obtained using hash function 1 and attaching the hash function used and the validation code to the colliding item. (h) the tenth participant is encoded without collision.

## ID lookup

To *Lookup* the ID of an existing participant an ID is computed using the default hash function. This ID is returned if there are no collisions associated with this ID. If there are collisions associated with this ID, the list of alternatives is assessed by first computing the alternative ID and validation codes for each entry in the list of hash functions. The alternative ID with a matching validation code is then returned as the participants ID.

With this scheme additional information is only stored for colliding items which represent a small fraction of the IDs in most cases. The additional information maintains anonymity as an attacker only knows alternative hash functions that are used, and not which ID they map to. The validation code does indeed reduce the anonymity of the item, but a phonebook attack will fail if the truncations are sufficiently large. Also, collisions are unlikely to occur before a large portion of participants have been added making it harder for an adversary to reverse engineer the resulting ID from the set of IDs.

Table 2 shows an example distribution of hash functions utilization when coding 100 participants with coding spaces of 1,000, 10,000, and 100,000, respectively. With smaller coding spaces more hash functions are needed than with larger coding spaces as 5 hash functions are needed to code 100 participants with a coding space of 1,000 while only 2 hash functions are needed to code 100 participants with a coding space of 100,000. In all cases, most of the coding is performed using the default hash function (hash-type 0). There is a theoretical chance that more hash functions are needed than what is provided. Given such a condition the algorithm detects that it is unable to find a free slot for that given participant. However, Table 2 suggests that such conditions are unlikely in practice as. It is also possible to extend the set of hash functions by adding shift and salt variations with the CRC32 function.

## Example

An example of the CANDIDATE algorithm is provided in Fig 3 where 10 arbitrary names are encoded into a space of 50 codes. In this example the phonetic step is omitted for simplicity. First the name "Rodman, David M" is encoded with ID = 16. Clearly, there are no collisions as this is the first item. The same holds for the subsequent six items. However, the eight item "Wetterau, John R." results in ID = 40 which results in a collision with the ID for "Mortensen, James K.". An alternative hash function (hash type = 1, reversing the name) results in the unique ID = 26, and the hash type (1) and the validation code 17 is attached to item no. 7. A similar situation occurs when adding the ninth item "Couper, Mick P." which results in ID = 18 which is already assigned to "Woodward, Mark". Therefore "Couper, Mick P." is coded with a different hash function (reversing the name) which gives the unique ID 30, and the coding type (1) and validation code of 47 is attached to the third item of "Woodward, Mark".

**Table 2. Distribution of hash functions used by CANDIDATE for encoding 100 participants with coding spaces of 1000, 10,000 and 100,000.**

| hash-type | Name | Frequency (%) | | |
| --- | --- | --- | --- | --- |
| | | *N* = 1,000 | *N* = 10,000 | *N* = 100,000 |
| 0 | djb2 | 95.0915% | 99.4947% | 99.9503% |
| 1 | CRC-32 | 4.5674% | 0.5011% | 0.0497% |
| 2 | djb2(reverse) | 0.3164% | 0.0041% | |
| 3 | djb2(shift-1) | 0.0228% | 0.0001% | |
| 4 | djb2(shift-2) | 0.0018% | | |
| 5 | djb2(shift-3) | 0.0001% | | |

Clearly, all the 10 items were successfully coded with two recoverable collisions. It is straightforward to look up the IDs of participants without collisions. To find the ID of an item with collision, say "Woodward, Mark" we first compute the ID with the default hash function and find 18. Since, 18 is associated with a collision we need to check the validation codes. Here, "Woodward, Mark" yields a validation code of 45 which does not match the entry 47. We can therefore assume that the valid ID is 18. If we instead looked up the ID of "Couper, Mick P." which also results in an ID of 18, we would find that its validation code matches the of 47 and we know the valid ID is obtained by applying hash-type 1, yielding the intended ID of 30. Fig 4 shows an implementation of the CANDIDATE tool that runs in the web-browser. The example shows **a** test user "Test A. User" is added into a space of 100 codes. The tool allocated ID = 11. A JSON object with the coding parameters for the sample is returned and needs to be stored. Subsequent lookups of the name with the given parameters will return ID = 11. The form input is also checked for invalid characters, and the researcher is warned about the anonymity limitation of a given study configuration. For example, if a study is to include 20 participants the population where these participants are recruited from need to comprise at least 500 individuals to achieve a minimum anonymity (k-anonymity = 5). This total sample population estimate is given by *k-anonymity* × *L*.

## Evaluation

The performance of the CANDIDATE tool was evaluated using simulations which allowed the tool to be exposed to many different scenarios. These simulations addressed two issues, namely the ability to successfully and uniquely link participants (integrity), and ability to preserve participants anonymity.

To test the tool a list of 103,472 researcher's names was taken from the dataset of a bibliometric study [50] adopted from [51] (see the GitHub repository). This list contains family names, in most cases first names, and in some cases initials. Each simulation was based on drawing a random sample of names from this master list and gradually adding these participants using the CANDIDATE tool. Two sets of sample sizes were used, namely 10 to 100 participants in steps of 10 representing small studies which are common in computer science [52] and 100 to 1,000 participants in steps of 100 representing medium to large studies which are more common within the health sciences. The simulations were repeated 10,000 times for each sample size and coding space configuration.

### Integrity

Figs 5–8 lists the results. The results in Figs 5 and 7 are representative of small studies with 10 to 100 participants encoded onto coding spaces with 100, 1,000 and 10,000 entries. With a coding space of 100 the IDs are in the range from 00 to 99 (two digits), with a space of 1,000 IDs are in the range from 000 to 999 (three digits) and for 10,000 IDs are in the range from 0000 to 9999 (four digits).

The results show that there was a non-zero probability of collisions for all the configurations ranging from 0.5% to 100%. However, the results show that the CANDIDATE tool successfully handled these collisions in most cases. With a coding space of 10,000 (four-digit IDs) all cases were handled successfully (up to 100 participants). With a coding space of 1,000 (three-digit IDs) there was less than 0.21% chance of the tool not being able to resolve the collisions, and with 10 and 20 participants all cases were handled successfully. The results show that coding up to 100 participants using a space of 100 entries (two-digit IDs) is more challenging. Coding 10 participants into a space of 100 entries gives a 0.1% chance of unrecoverable collisions, 20 participants can be coded with 0.91% chance of unresolvable collisions, while

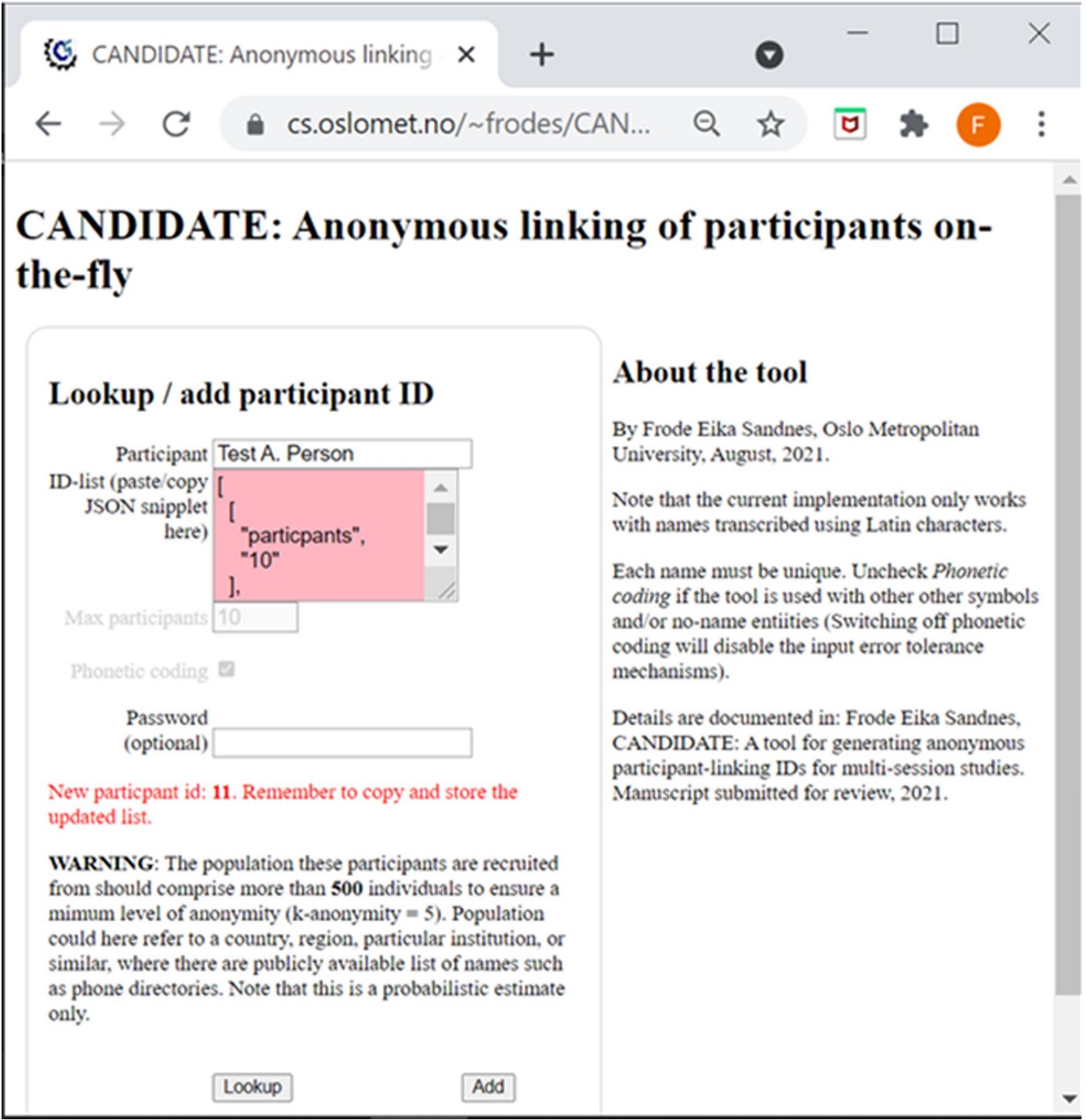

**Fig 4. A screenshot of the CANDIDATE tool implementation.**

with 30 participants there is nearly a 3% chance of collision that cannot be automatically resolved.

Figs 6 and 8 list simulation results for medium to large studies with 100 to 1,000 participants mapped onto a coding space of 10,000 (four-digit IDs) and 100,000 entries (five-digit

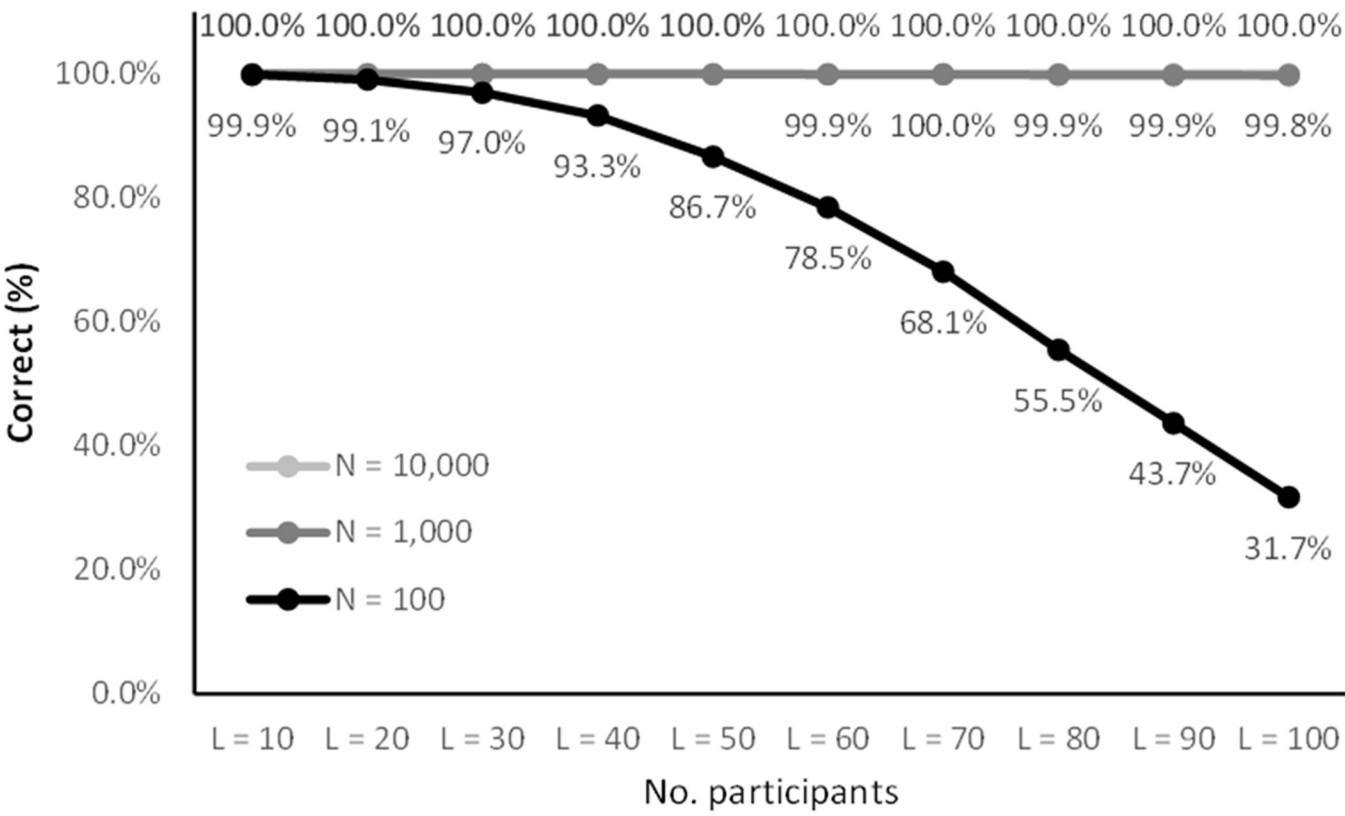

**Fig 5. Encoding success rates for small samples ($N \leq 100$).**

IDs). With the 100,000 space there were no cases that resulted in unresolved collisions, while with 10,000 entries the up to 1,000 participants were coded with less than 0.26% chance of unresolved collisions. Up to 200 participants were coded without any unresolved collisions.

## Anonymity

To assess the anonymity of the CANDIDATE tool one successful encoding was selected from each of the conditions shown in Figs 5 and 7. Each encoding was subjected to a phonebook attack using the full list of 103,472 names. The number of hits per ID were recorded as well as the number of names that did not result in a valid hit and the amount of coding space without hits. It was assumed that an attacker had access to the list of valid IDs and the coding tables. The results are shown in Figs 9–11.

Fig 9 shows a log-log plot of the k-anonymity with coding spaces of 100, 1,000, 10,000, and 100,000 slots, respectively. With a coding space of 100 items (two-digit IDs) the minimum number of hits per slot was 818 (mean = 1,035). This means that no items can be uniquely identified through a phonebook attack. This k-anonymity is much higher than the recommended minimum of 5. With a coding space of 1,000 items (three-digit IDs) the minimum number of items per slot was 71 (mean = 103), which is also high. With a coding space of 10,000 (four-digit IDs) the smallest number of hits per ID is 1. This means that it was not possible to hide the identity of certain individuals, and that the anonymity of these individuals cannot be guaranteed. However, for most individuals this configuration provides sufficient anonymity as the mean number of hits per ID is 10.35 which is above the recommended limit of 5.

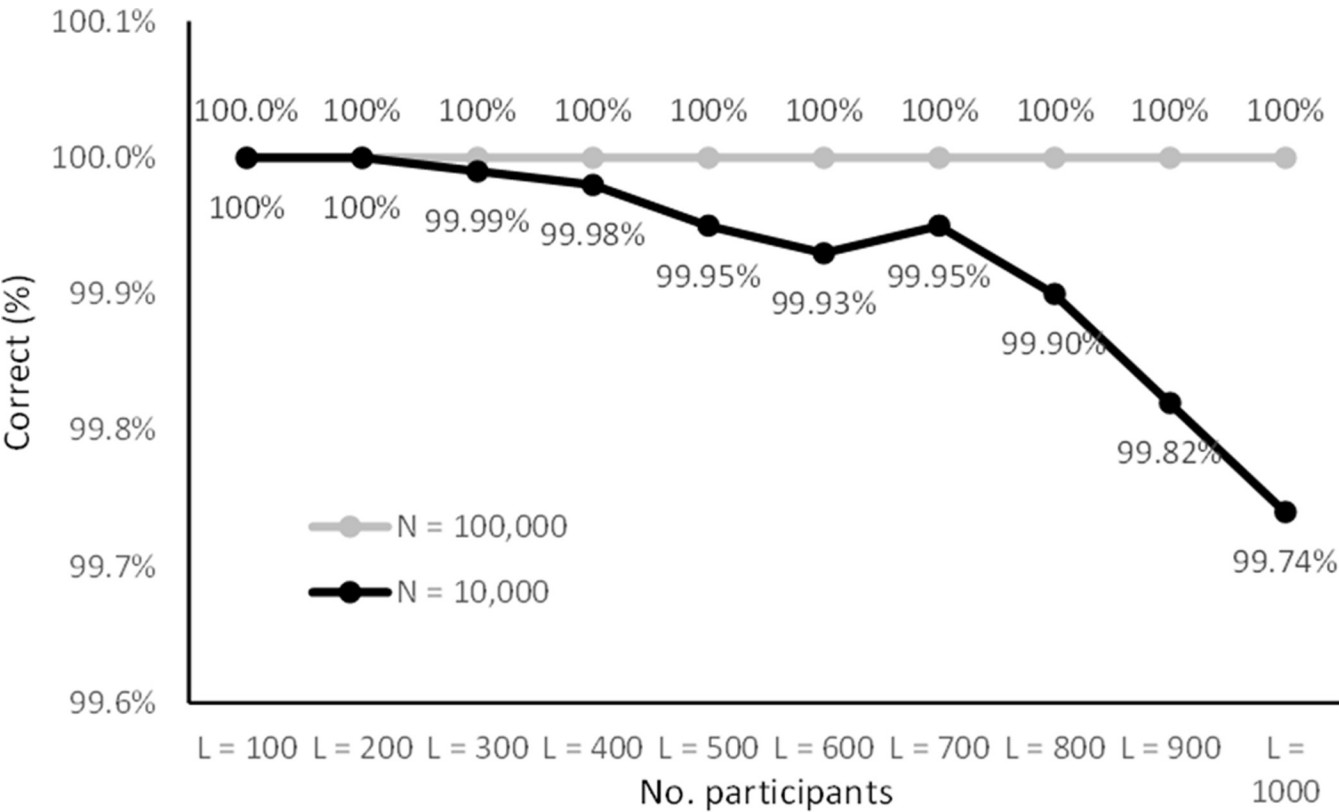

**Fig 6. Encoding success rates for larger samples ($100 \leq N \leq 1000$).** Note that the y-axis starts at 99.6% to show the small variations.

When considering the ratio of phonebook items that can be confirmed as not being part of the study as they result in unused IDs, the results show that this ratio matches the ratio of participants to coding space size. That is, when 10 participants are coded with a coding space of 100, the 90% of the phonebook entries can be rejected. If 100 participants are coded with a coding space of 100 (two-digit IDs), none of the phonebook entries can be rejected. With coding spaces of 1,000 and 10,000 the rejection rates ranged from 90% to 99% and 99.0% to 99.9%, respectively. Although most of the phonebook entries could be rejected the items that were not rejected made up a larger set of candidates than the number of participants. The results show that the phonebook attack reached all the items in the coding space for all conditions.

With 10,000 slots (four-digit IDs) the mean number of hits per ID is 10.34 which is above the limit of acceptable anonymity. However, the smallest number of items per ID was 1, which indicates that anonymity cannot be guaranteed for some individuals. With 100,000 slots (five-digit IDs) there is a mean of 1.03 hits per slot which cannot be considered anonymous. Clearly, the k-anonymity is related to the size of the coding space in relation to the total population and not the number of participants.

Fig 11 shows that with a coding space of five digits only between 0.2% and 2% of the names in the phonebook resulted in valid IDs and could therefore be classified as not being part of the sample. The results also showed that with this configuration more than a third of the slots in the coding space remains unused.

Note that the anonymity results depend on the size of the phonebook. With smaller phonebooks (small populations) the anonymity will be lower, and larger phonebooks (large populations) will obviously result in higher anonymity. The size of the phonebook used herein is

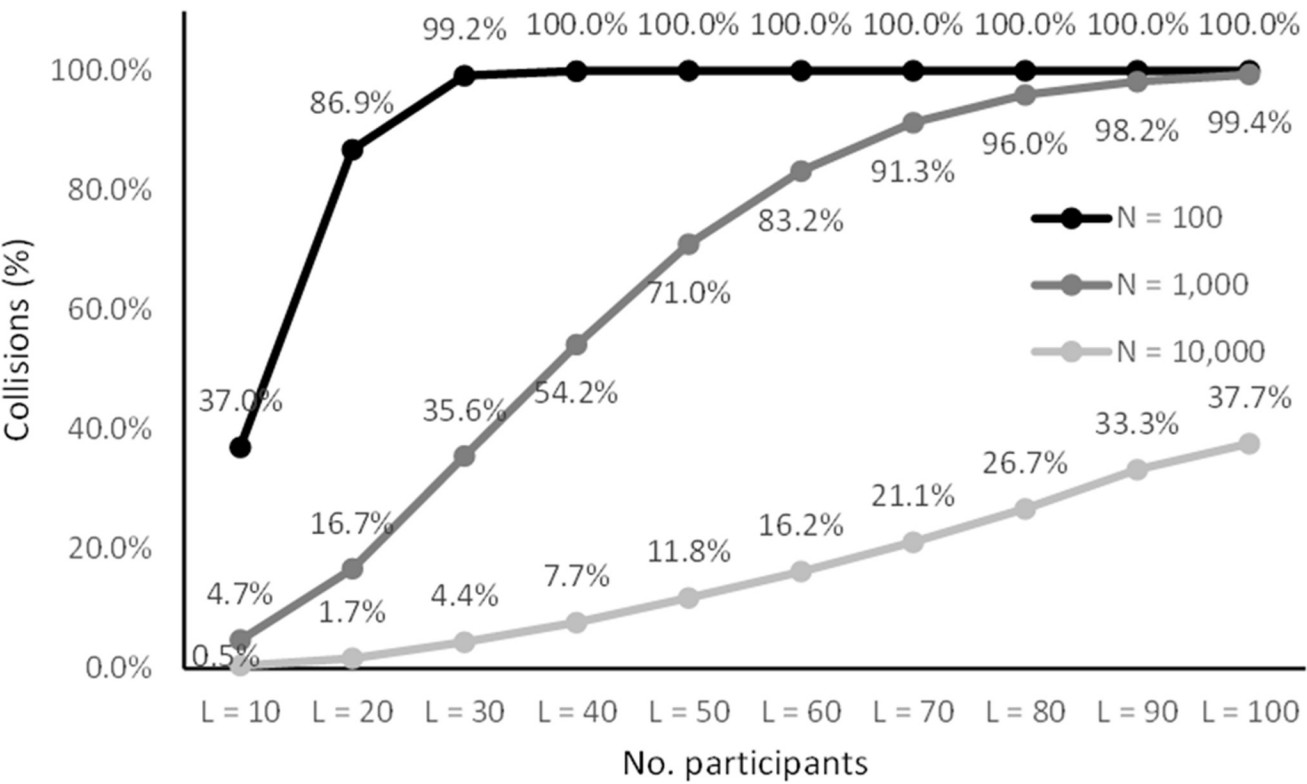

**Fig 7. Collision rates for small samples ($N \leq 100$).**

comparable to a small country such as the U.S Virgin Islands (a population of 104,425 in 2020).

## Discussion

The results confirm the trade-off between the ability to recover from collisions and the degree of anonymity of the resulting IDs. By increasing the ID-lengths the probability of unresolved collisions is reduced, while at the same time the anonymity is weakened. Short IDs yield high anonymity while there is a higher probability of encoding errors due to unresolved collisions.

The results suggest that a reasonable compromise is achieved if the coding space is ten times the number of participants, that is, with $L$ participants the coding space should be $N = 10 \times L$. For example, coding 100 participants using three-digit IDs (a space of 1,000 slots) there is a 99.79% chance that there will be no unresolved collisions. In plain terms, this means that if one conducted 500 different research projects there would be one research project with an unresolved collision. In practical terms, the effective chance of experiencing unresolved collisions is low.

What are the consequences of unresolved collisions? Unresolved collisions result in erroneous IDs. The corresponding observations of the erroneously identified participant would therefore be incorrectly linked. However, the researcher may detect the error as a duplicate data entry for an ID and may be able to resolve the issue manually.

Imagine one ends up with one erroneous ID when coding 100 participants in a between-groups experiment comprising two distinct groups of participants. Furthermore, this participant is incorrectly mapped to the incorrect group. When analysing the data, the researcher

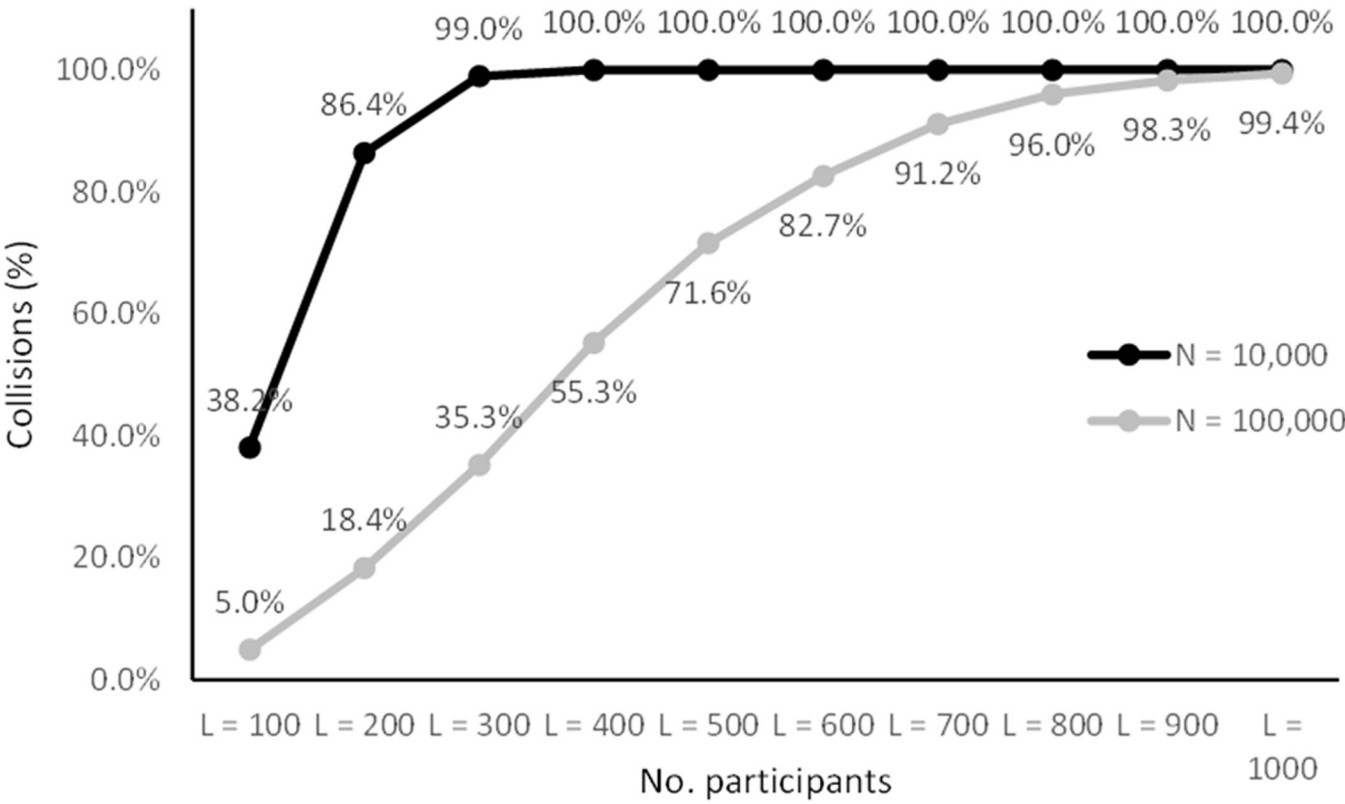

**Fig 8. Collision rates for larger samples (100 ≤ N ≤ 1000).**

will then notice that one participant is associated with a double set of observations and another and that the number of unique ID is less than the number of participants. One option is to discard this set of data. Two of the 100 observations will therefore be removed. Either one set of data for each of the two groups are discarded, or two sets of data for one group are discarded. However, statistical procedures should be sufficiently robust to handle any imbalance and without affecting the overall conclusions. Exceptions may of course occur if the results rest on the borderline of statistical significance. In fact, statistical procedures are likely to handle several incorrectly mapped items.

In conclusion, we argue that the benefits of preserving participants' privacy, perceptually simple and short ID codes, and simple administration of research studies outweigh the small risks associated with unresolved collisions.

The anonymity provided by CANDIDATE would be weaker with large datasets, such as large biobank studies which can include data about several million individuals [53]. This is because more individuals would have a unique ID that would allow them to be uniquely identified. Other anonymity mechanisms should therefore be used with such large-scale studies. However, if a study involves a smaller subset of such a dataset CANDIDATE may be used. This is because the CANDIDATE coding would be specific to that subset, while a portion of other individuals in the superset will yield false positives. An adversary can therefore not be certain if a match is true or false. It must also be noted that the proposed procedure does not relieve the researchers from the responsibility of reflecting over the ethics, privacy, and implications of a research study. Even though an anonymous procedure is used it does not mean that the stored data are anonymous. For instance, pieces of demographic information that in

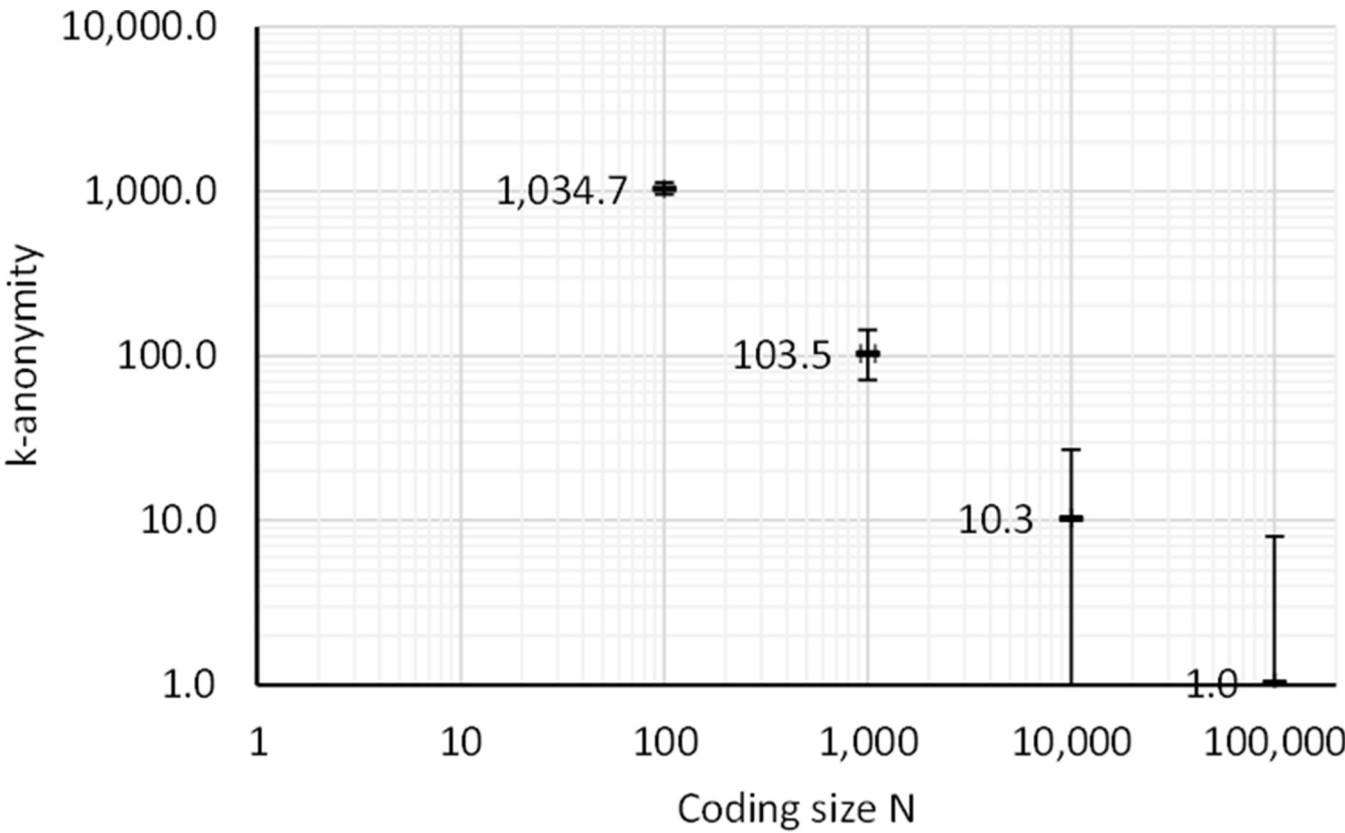

**Fig 9. Log-log plot of mean anonymity with coding spaces of 100 (two-digit IDs), 1,000 (three-digit IDs), 10,000 slots (four-digit IDs), and 100,000 (five-digit IDs) with a phonebook of 103,472 names.** Error bars indicate the minimum and maximum anonymity.

isolation do not reveal the identity of a participant, may reveal the identity when these pieces of information are combined. Moreover, qualitative data may contain information that reveals someone's identity. The content of the data stored also needs to be carefully considered during the design of studies and experiments.

An important point is that the participants' identities need to be known by the researcher at some point, when the participant is invited to participate, or when a participant is returning to a subsequent session. Even email correspondence with participants in the researcher's email account could be considered a name list, and email accounts have been found to be vulnerable to security breaches. Clearly, if the list of participants is written down and stored the linking table problem persists. If an attacker gets hold of such a list of names, the attacker can with certainty know who participated in a study and subsequently find the IDs of the participants and know which data belong to which person. The researcher should therefore avoid keeping a list of participants, email correspondence and similar items. For small studies it is feasible for a researcher to memorize who the participants are, but this is not practical with larger studies. Another approach is to recruit participants in person (in some physical location such as a street, shopping centre, workplace, school, or hospital) and then make an (scheduled or unscheduled) appointment to turn up for one or more sessions. The researcher only needs to record the ID (and possibly time). The participant is responsible for turning up for a first, or subsequent, session. They then produce their name so that the ID can be found. Clearly, there is a probability that some participants will forget, but that may be a justified compromise to achieve anonymity.

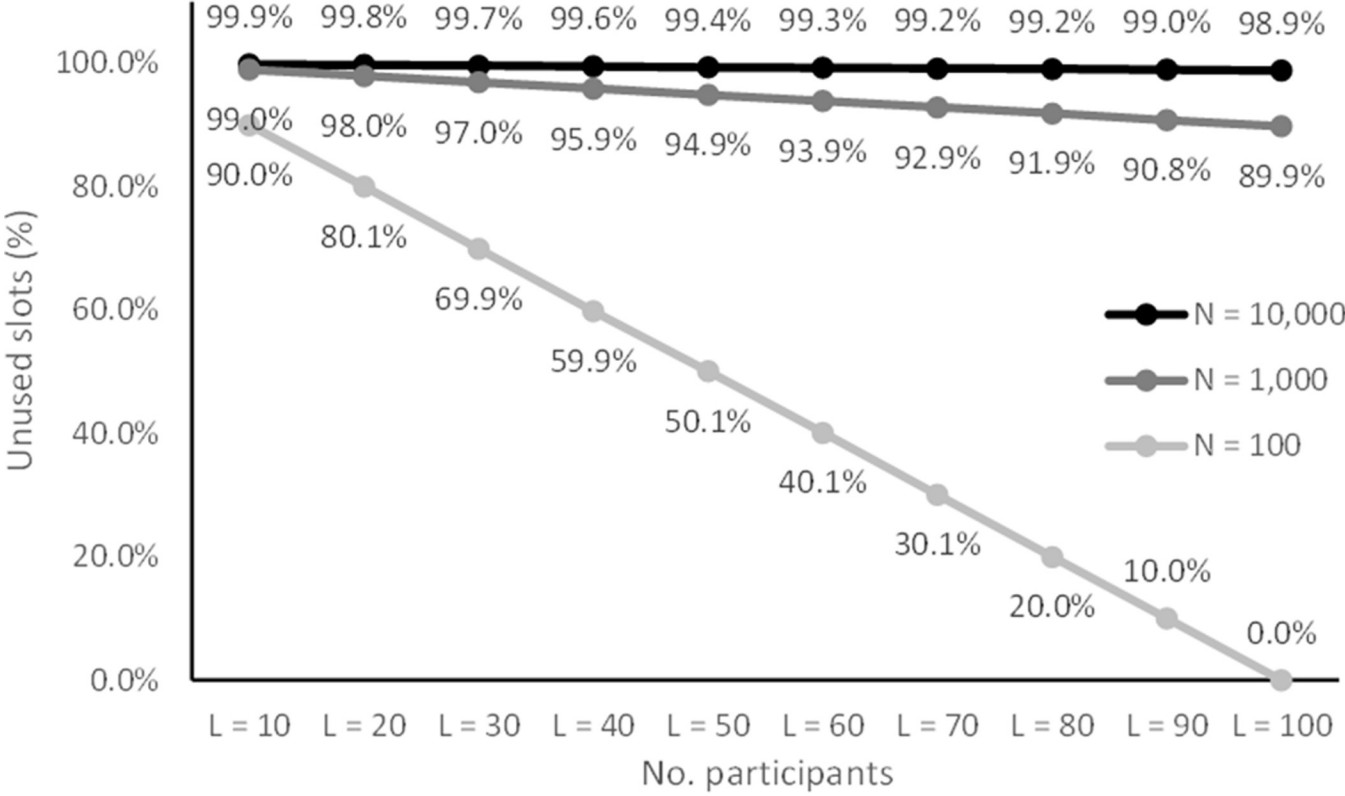

**Fig 10. Percentage of unused ID slots with small sample sizes.** This indicates the portion of phonebook entries that can be discarded as non-participants during an attack.

The CANDIDATE tool does not facilitate data from one study to be linked to data in other studies as the IDs are only valid for the set of participants within a given sample. This may be viewed as a limitation of the approach and an inconvenience to researchers. However, one may also argue that this in fact is a beneficial feature as CANDIDATE supports the ethical principle that data should only be collected for a specific and well-justified purpose and for that purpose only as the reuse of sensitive data in new contexts is associated with several problematic issues related to trust and privacy [54, 55]. Privacy of participants and guarantees that the powers of the consent given are adhered to trumps convenience for researchers.

Although several mechanisms are employed to reduce the impact of input errors, there are several types of input errors that cannot be handled automatically. For instance, the tool will be unable to find a match if a participant's middle name is inconsistently included or omitted on subsequent occasions. Researchers therefore need to be as accurate as possible when inputting participant information. Also, the input error tolerance mechanisms assume a Latin character representation. Other languages such as Chinese, Russian, or Arabic may require other error tolerance mechanisms.

This study focused on coding participants by name and with this scheme the researcher needs to ensure that all the names are unique. With large datasets there is a probability that some participants share the same name. With larger sample sizes researchers may achieve uniqueness by concatenating additional information such as date of birth (day of the month, month, year, or combinations thereof) to the name. On the downside, incorporating additional information will increase the complexity of administering the study.

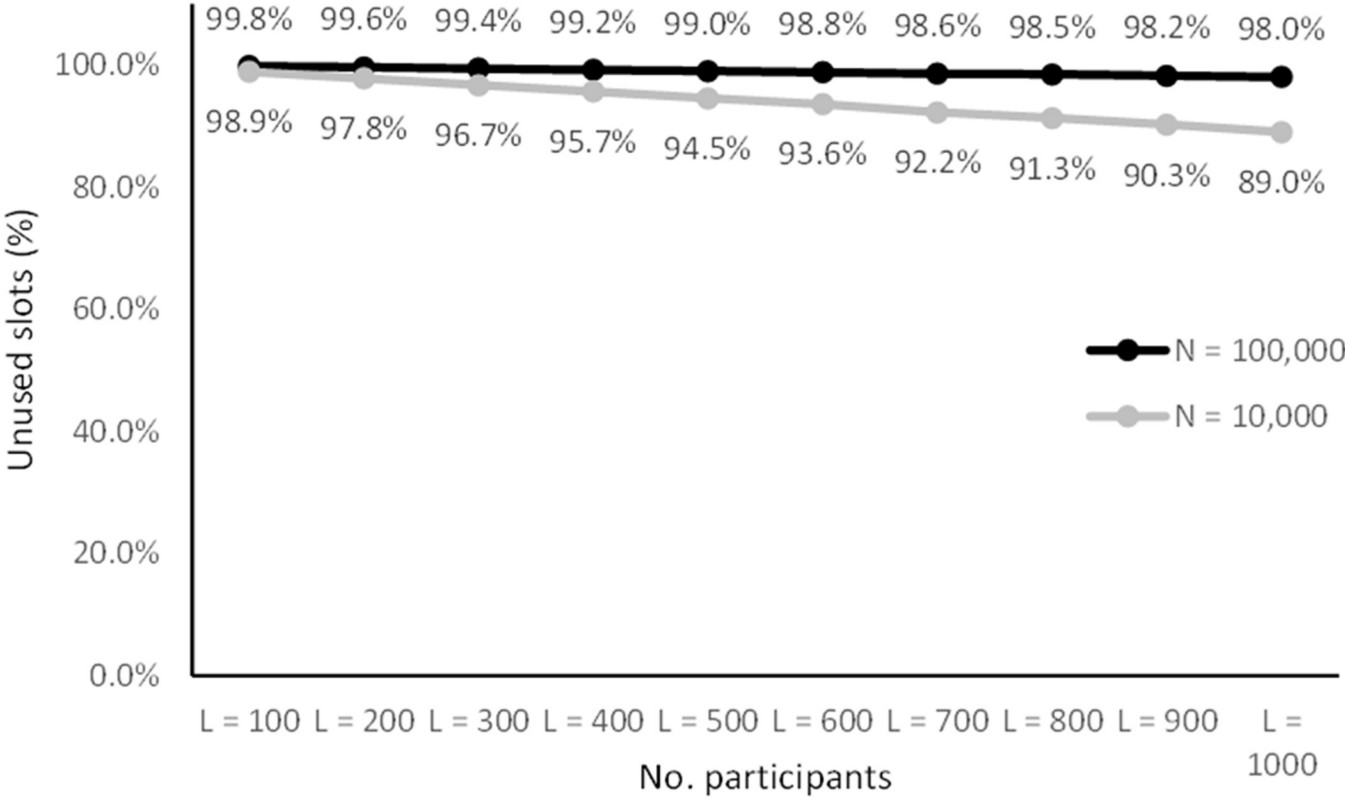

**Fig 11. Percentage of unused ID slots with large sample sizes.** This indicates the portion of phonebook entries that can be discarded as non-participants during an attack.

Not explored in this study is the coding of non-name representations such as genetic or biometric information. To use such coding types, the representation of an individual must be consistent and identical on every instance. If the instances of one individual vary CANDIDATE will not be able to generate consistent codes. Biometric matching is often performed using a set of approximate parameters where matches are determined using some distance functions, such as degree of matching in fingerprint recognition [56]. Two biometric measurements of the same individual will rarely be identical and therefore cannot be coded using CANDIDATE.

## Conclusions

The CANDIDATE tool for flexible and anonymous linking of participants was presented. Evaluations show that the tool with appropriate parameters can successfully assign unique and anonymous IDs to participants with a very low probability of obtaining unresolvable ID collisions. Experiments showed that if the space of IDs is about ten times the number of anticipated participants, one achieves a good balance of integrity and anonymity. In the very unlikely situation that incorrect IDs are generated due to collisions the robustness of statistical testing should ensure that the overall conclusions are not affected (false positives or false negatives) if one employs a hypothesis testing paradigm, or similar, CANDIDATE holds potential for simplifying the administration of multi-session studies. More researchers may be encouraged to follow participants over time to collect solid empirical data that allow researchers to draw reliable conclusions. An implementation of the CANDIDATE procedure that can be used locally in a web-browser (https://www.cs.oslomet.no/~frodes/CANDIDATE/) has been made

available to the research communities as well as the simulation code of researchers who want to further develop the procedure (https://github.com/frode-sandnes/CANDIDATE/).

## Supporting information

**S1 File.**
(TXT)

## Author Contributions

**Conceptualization:** Frode Eika Sandnes.

**Data curation:** Frode Eika Sandnes.

**Formal analysis:** Frode Eika Sandnes.

**Investigation:** Frode Eika Sandnes.

**Methodology:** Frode Eika Sandnes.

**Project administration:** Frode Eika Sandnes.

**Software:** Frode Eika Sandnes.

**Validation:** Frode Eika Sandnes.

**Writing – original draft:** Frode Eika Sandnes.

**Writing – review & editing:** Frode Eika Sandnes.

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
