## [Decision Letter · Decision Letter 0]

27 Jul 2021

PONE-D-21-20957

CANDIDATE: A tool for generating anonymous participant-linking IDs for multi-session studies

PLOS ONE

Dear Dr. Sandnes,

Thank you for submitting your manuscript to PLOS ONE. After careful consideration, we feel that it has merit but does not fully meet PLOS ONE’s publication criteria as it currently stands. Therefore, we invite you to submit a revised version of the manuscript that addresses the points raised during the review process.

We look forward to receiving your revised manuscript.

Kind regards,

Anandakumar Haldorai, PhD

Academic Editor

PLOS ONE

Journal Requirements:

 “This study was not funded. The funders had no role in study design, data collection and analysis, decision to publish, or preparation of the manuscript.”

Additional Editor Comments:

Please carefully address the issues raised in the comments and, up front in your revised paper. Your revised paper will be sent to the same reviewers, as well as possibly new reviewers, for evaluation.

Make sure the Abstract briefly describes the paper as it is used in abstracting and citation services. Keep the Abstract between 200 words. Do not use any references in the Abstract.

Spell out each acronym the first time used in the body of the paper. Spell out acronyms in the Abstract only if used there.

Include a list of six to ten key words after the Abstract.

You may ignore any suggestion of including self-references by reviewers if not applicable.

Include a paragraph at the end of the Introduction describing the organization of the paper.

Make sure that the Conclusion briefly summarizes the results of the paper it should not repeat phrases from the Introduction. Keep the Conclusion to about 300 words. Do not use any references or acronyms in the Conclusion.

Make sure all figures and tables are referred to in the body of the paper.

Properly follow PLOS ONE reference style in both reference and citation sections.

It is recommended to use a professional native English-speaking editor. Papers with less than excellent English will not be published even if technically perfect.

Reviewers' comments:

Reviewer's Responses to Questions

**Comments to the Author**

1. Is the manuscript technically sound, and do the data support the conclusions?

Reviewer #1: Partly

Reviewer #2: Yes

2. Has the statistical analysis been performed appropriately and rigorously? 

Reviewer #1: Yes

Reviewer #2: Yes

3. Have the authors made all data underlying the findings in their manuscript fully available?

Reviewer #1: Yes

Reviewer #2: Yes

4. Is the manuscript presented in an intelligible fashion and written in standard English?

Reviewer #1: Yes

Reviewer #2: Yes

5. Review Comments to the Author

Reviewer #1: See attachment .

Reviewer #2: The manuscript by Sandes describes the CANDIDATE tool, a computational method for anonymizing participant identities in research studies. Due to real and perceived privacy concerns, as well as regulations such as the GDPR, there remains a need for simple, robust tools for maintaining the privacy of individuals in research studies, while enabling researchers to potentially link information from the same individual across longitudinal studies or multiple studies.

Although I work in a different field and thus I am not familiar with most of the existing tools, I believe the manuscript did a good job of explaining the challenges and existing tools in the field. In particular, the computational modeling of different study sizes and scenarios is necessary to understand the interplay between study size, anonymity, collision rate (same ID assigned to multiple individuals) vulnerability to “phonebook attacks” to decipher identities, and the ID space (length of ID code.) The main strength of the manuscript is the clear description of the tool, and the easy-to-understand examples. In particular, I appreciated the recommendation that the ID pool be approximately 10x the size of the participant list (e.g., encoding 100 participants with a 3-digit number.) A weakness of the manuscript is that the body or discussion could have included some information about extensions of the tool (for example, whether a tool like this could be adapted to work with numerical, or genetic, identity information instead of names.) Also depending on space, some of the tables as have been easier to interpret as graphs of curves. Overall the CANDIDATE tool appears simple to use and useful, and I recommend publication of the manuscript.

Specific issues:

I think the manuscript could benefit from putting the work into a broader context. A graph or two in place of some tables would make it easier to quickly interpret the study.

a. The discussion of the need for privacy is excellent, however, the current study is limited to encoding of individuals with roman alphabet names encoded by 26 characters. It is possible that a deeper discussion of the Soundex algorithm could rectify this, however, it is unclear whether the Soundex algorithm can deal with Chinese or Russian names, or whether these would need to be romanized. Alternatively, are there other tools that can sanitize name information or make them more robust, that could be used in place of Soundex tool?

b. It would be interesting to learn if the tool could be enhanced or even used with participant IDs that were coded with a combination of numbers and letters, or a combination of names and genetic or biometric information, for example. Alternatively, can or should the CANDIDATE tool be adapted to use an alphanumeric ID space, instead of a strictly numeric ID space as in the examples. While this may be speculative and beyond the scope of the study, there is a clear future need for robust identification of individuals, whilst preserving their privacy.

c. The modeling appeared robust for groups of 10-1000 individuals, but it is unclear whether this tool could be used on a larger scales, such as encoding information for 10’s of millions of individuals, or a large nation such as the UK. If this is not possible, it could be useful to provide context of whether CANDIDATE could be useful in subsets of those large biobank studies.

d. I am not familiar with the details of different hash functions. The manuscript did a good job of explaining the goal of the hash, to provide one-way encoding, but it would be nice to have an explanation of different hash functions, and whether most are equivalent, and what are the tradeoffs of using multiple hashes VS single hashes.

Finally, while the language is clear, the manuscript could benefit from copyediting for language and spelling.

6. PLOS authors have the option to publish the peer review history of their article (what does this mean?). If published, this will include your full peer review and any attached files.

Reviewer #1: **Yes: **Mario Lorenz

Reviewer #2: No

---

## [Author Response · Author response to Decision Letter 0]

14 Aug 2021

RESPONSE TO REVIEWERS’ COMMENTS MEMO: PONE-D-21-20957

Reviewer #1 comments:

REVIEWER #1(pdf): 1. I tried the code from GitHub. Using the htm file. There seems to be bug when entering the same input multiple times. I can at least produce 2 different id’s. Further, If I add Symbols, Numbers letter to an existing string of solely latin-letters I receive the same ID, Only by adding another latin-character the ID changes. Although, this is no evaluation criteria for the paper it makes me wonder if these are just bugs or conceptual errors? 

[RESPONSE]: I am very grateful that the reviewer set aside time to try the tool. What the reviewer observed is indeed correct and it is intentional. To encode input with digits and other symbols the user needs to uncheck the “phonetic coding” checkbox in the form (checked as default). When using phonetic coding (for error tolerance) the input is sanitized. The tool is not capable of handling identical names. If adding the same name twice the tool will return two codes, but the user will only get one of these codes when looking up the same name. Since the names are not stored it is not possible for the tool to know if a name is repeated or not. A comment was added to the tool to make the user aware of both the name uniqueness requirement and the purpose of the phonetic coding. Also, a comment about the possibility of disabling the phonetic coding step was also added to the Encoding participants Section.

REVIEWER #1 (pdf): 2. Non-latin letters are not treated as letters. A combination of Non-latin letters and latin letters leads to an reset of the application. I tried with Arabic letters. 

[RESPONSE]: Yes, this observation is correct. The current implementation of the tool does not support non-Latin characters. Names in other languages (such as Arabic) need to be transcribed using Latin characters. The section on CANDIDATE has been adjusted to explain this limitation and a paragraph was added to the Discussion section which describes possible support for other languages. Information about this limitation was also added to the tool. 

REVIEWER #1 (pdf): 3. Further from an UX POV I wouldn’t let the user chose the ration as it is an error source. Better automatically set it to a reasonable value based on the max participants input. 

[RESPONSE]: This is a valid point. I decided to follow the advice of the reviewer and have disabled the option to change the ratio (I set it to 10 to reflect the recommendations resulting from the experimentations documented in the manuscript). Advanced users who need a different ratio may manually alter this in the source.

REVIEWER #1 (pdf): 4. “The privacy and safety of participants is of utmost importance in research that involves people. Privacy is also regulated by legislation such as the General Data Protection Regulations (GDPR) which applies in the European Union.” ��For research the more important and global codex for experiments is the Declaration of Helsinki. Pls refer to it. 

[RESPONSE]: This is an excellent suggestion. A reference to the Helsinki Declaration was added in the sentence before the mentioning of GDPR.

REVIEWER #1 (pdf): 5. Pls explain what a salt is on its first occurrence in the introduction. 

[RESPONSE]: An explanation of salt was added as suggested.

REVIEWER #1 (pdf): 6. The first 2 paragraphs in related work are motivational and should be integrated in the introduction. 

[RESPONSE]: The two paragraphs were moved from the Related work section and integrated in the Introduction section.

REVIEWER #1 (pdf): 7. “Experimentation showed that a suitable compromise between integrity and anonymity is achieved when N is ten times the anticipated number of participants” �This is anticipation of the Evaluation results. Pls consider removing or explicitly referring to the evaluation section. 

[RESPONSE]: I agree with the reviewer. This sentence was removed.

REVIEWER #1 (pdf): 8. It is misleading, that the author mostly speaks of the name as an input variable, although any String could be used. When read superficially one could be mislead that the tool is not suitable for larger studies where it is likely to have multiple participant with the same name. I would strongly advise to scatter the information that any unique string generated from easy accessible information can be used as an input. That’s said: Pls add this as possible limitation, as it is up to the experimenters ensure such unique input strings. 

[RESPONSE]: The reviewer is right. The following paragraph outlining this limitation was therefore added to the Discussion section: “This study focused on coding participants by name and with this scheme the experimenter needs to ensure that all the names are unique. With large datasets there is a probability that some participants share the same name. With larger sample sizes experimenters may achieve uniqueness by concatenating additional information such as date of birth (day of the month, month, year, or combinations thereof) to the name. On the downside, incorporating additional information will increase the complexity of administering the study

REVIEWER #1 (pdf): 9. Line 191 ‘L’ and ‘ID-List’ are not previously defined 

[RESPONSE]: Both L and ID-List have now been defined in the pseudo code.

REVIEWER #1 (pdf): 10. 191 hash-typefree = Find-free-slot(name, L, ID-List) ��Isn’t this bound to often return a value >9? Then switch in hash function hardly make any sense. This leads me to the point that I do not understand why there are 10 different digest generating function in the hash function. 

[RESPONSE]: The find-free-slot will usually return a value less than 9 (in most practical cases). But, it can return a value of 9 or more. The pseudo-code has been adjusted to make the intended meaning clearer. The last switch statement in the hash function will trigger if hash-code is equal to 9 or greater, that is, 10, 11, 12 … etc (the ≥ symbol now used). The returned hash is then the djb2-hash with a salt added. The salt is taken from an array of salts with index given by the hash-code (a declaration of the salt array was added). The different types of digests are therefore simply bound by the number of salts in the salt array. 

The reason why the hash function can return many different types of digests (10+) is to be able handle collisions. I.e., if two different inputs result in the same digest with one hash function they will (usually) result in two different (non-colliding) digests with two different hash-functions. The different digest thus facilitates the computation of alternative-IDs. Then, which hash function to use is stored with the ID-original item. A new figure was added to help illustrate the use of different hash functions to handle collisions.

REVIEWER #1 (pdf): 11. I think Add-Function (Line 186) could be simplified by inverting the check in the if-statement. 

186 Add(name, N) 187 IDoriginal = Encode(name, N, hash-typedefault) 188 IF IDoriginal in ID-list 189 hash-typefree = Find-free-slot(name, L, ID-List) 192 IDoriginal = Encode(name, N, hash-typefree) 193 validation-code = Encode(name, N, hash-typefree + hash-typeoffset) 194 ATTACH (hash-typefree, validation-code) TO IDoriginal 

195 Add IDoriginal to ID-list 

[RESPONSE]: This is a good suggestion. The Add-function was simplified.

REVIEWER #1 (pdf): 12. Line 194 souldn’t it be IDalternative 

[RESPONSE]: No, the statement is correct. It is because different names that result in colliding IDs (ID-original) will first “land on this slot”. The fact that there are already one or more validation-codes assigned to this ID-original means that a different ID (ID-alternative) needs to be created and assigned. During lookup we need to inspect the validation-codes to identify the matching one and the correct hash can be applied to find the ID-alternative. This is explained in the passage starting with “When adding a new participant, we first compute….”

REVIEWER #1 (pdf): 13. In add and lookup function ‘=’ is used the other function use ‘:=’ pls unify 

[RESPONSE]: Thank you for spotting this inconsistency. All assignments have been unified to “:=”.

REVIEWER #1 (pdf): 14. “199 FOR EACH (hash-type, validation-code) ATTACHED TO IDoriginal” parameters in FOR EACH not previously defined 

[RESPONSE]: The hash-type and validation-code in the FOR EACH are declared in the lines above (in the same IF block)

REVIEWER #1 (pdf): 15. It is bad programming habit to have more than one return statement. Pls revise the code of Lookup function. I find it further strange, that this function returns a newly generated code in case the FOR EACH runs without running into the THAN condition of the IF statement. Pls explain. 

[RESPONSE]: The code has been revised with just one return statement.

REVIEWER #1 (pdf): 16. I understand why the Sanitize method removes these characters, however maybe an Experimenter relies on these removed characters in order to create unique input strings. In the input text field I would check for the removed characters and make them invalid input. 

[RESPONSE]: This is an excellent suggestion. An input check was added to the tool that now only allows variations on the Latin alphabet, numbers (in case of phone numbers), @ and dot (in case of e-mails), and hyphen (as used in some connected names). The manuscript was also updated to reflect this change.

REVIEWER #1 (pdf): 17. Code of Soundex function is missing 

[RESPONSE]: I did not include a detailed description of Soundex to save space as it is quite a well-known algorithm. To make this clearer to the reader I added the statement “Since Soundex is well-documented (see for instance [11, 30-35]), with many available implementations, it is not described in detail herein.”. However, if the reviewer insists, I would of course be happy to expand the text with a detailed description of Soundex.

REVIEWER #1 (pdf): 18. “245 (without the four-character length restrictions)” �Out of nowhere statement. What is this 4 char restriction? 

[RESPONSE]: I agree that this sentence appears very cryptic when viewed out of context. It was therefore replaced with an example and an explanation, namely: “For example, “Christian” would be coded as C6235, i.e., the first letter (C), 6 for the r-sound, 2 for the c/g,/j,/k/q/s/x/z-sounds, 3 for the t/d-sounds and 5 for the m/n-sounds. Note that the full-length encoding is used, which differs from the original Soundex algorithm which only returns the first four characters (C623).“.

REVIEWER #1 (pdf): 19. Pseudo code of CONFIDENCE procedure is missing 

[RESPONSE]: This is a typo. It was corrected to CANDIDATE.

REVIEWER #1 (pdf): 20. References to the hash coding algorithms are missing (CRC32, etc.) 

[RESPONSE]: References to the detailed descriptions of the two algorithms were added to the manuscript. 

REVIEWER #1 (pdf): 21. “The CANDIDATE anonymisation tool“ section needs more structuring. Pls add meaningful subheading 

[RESPONSE]: Several subheadings were introduced to help guide the reader.

REVIEWER #1 (pdf): 22. General Comment: Pls add flow charts or sufficient UML diagrams for all algorithms described in section “The CANDIDATE anonymisation tool“. It would allow for drastically shorten text, less repetition and far easier understanding. 

[RESPONSE]: A diagram illustrating the essence of the algorithm, namely the collision handling using multiple hash functions was added to the revised manuscript (as this is the core of the approach). A diagram illustrating the generation of the different hash functions was also added. An attempt was made to make it understandable for a wider readership as knowledge and experience is required to read and interpret UML diagrams.

REVIEWER #1 (pdf): 23. A general comment: If the input string can be anything, then functions like Sanitize and Soundex are a bit meaningless, as they are intended to deal with names. They also seem to only work on latin-letter input. Also in the further description of CANDIDATE everything seems to be directed on handling names as input strings, although they are not unique and would be an unfavorable choice to take. 

[RESPONSE]: It is correct that Sanitize and Soundex only apply to name representations. From an information theoretic perspective there are better choices than just names as correctly pointed out by the reviewer, but from a practical experimenter’s perspective names are more acceptable as some participants are uncomfortable disclosing private information about themselves (including birth of dates). If the participation of a study is perceived as too “intrusive” participants may withdraw. Therefore, the emphasis is on names. Several statements have been revised and added in the revised manuscript to make this point clearer.

REVIEWER #1 (pdf): 24. Line 274-290: In the pseudo code of the add function there is no loop. So I am not comprehending where “we search through an array of hash functions” should occur? 

[RESPONSE]: The “loop” is inside Find-free-slot(..) in the add function. To make this clearer “we search through …” was replaced with “Find-free-slot searches through …”. 

REVIEWER #1 (pdf): 25. 296-311 are mostly providing information already given. 

[RESPONSE]: The passages have been revised to avoid redundant information.

REVIEWER #1 (pdf): 26. General Comment: Pls add the name list used to increase replicability. Should be as a supplement. 

[RESPONSE]: Done! The full list of names used for the experiments have been uploaded on the project GitHub page (link in the manuscript).

REVIEWER #1 (pdf): 27. Table 4/5/6/7. Pls type out all numbers 

[RESPONSE]: As suggested by Reviewer #2 the tables were replaced by charts to simplify interpretation and comprehension. All the repeated numbers are included in the chart (i.e. the repetition marks in the tables are no longer an issue.

REVIEWER #1 (pdf): 28. Table 8: why is ‘unused’ column not present for N=10,000 and in Table7? 

[RESPONSE]: Unused was not listed because the entire coding space was used (100%), hence it did not seem relevant to list it. To make this clear it is now explicitly stated in the text. Note that the content of Table 7 is now replaced by charts.

REVIEWER #1 (pdf): 29. Table7/8: How can you explain, that min/mean values are the same in all conditions except for N=100? 

[RESPONSE]: This is because the min and mean k-anonymity depends on the size of the coding space in relation to the total population (phonebook) and not the number of participants. A statement was added to make this explicit in the text.

REVIEWER #1 (pdf): 30. General Question regarding Evaluation: From what I understood from the explanation of the CANDIDATE algorithm, Anonymity and Encoding success greatly depends on the truncated hash value. In this case, this parameter is not given in the Evaluation section, and it is not varied to evaluate its influence. 

[RESPONSE]: Yes, the anonymity and encoding success depends on the level of truncation. The level of truncation (coding space) is denoted by N and this parameter is listed in the results.

REVIEWER #1 (pdf): 31. “With 10,000 slots (four-digit IDs) the smallest number of items per ID was 1, which indicates no anonymity. However, the mean number of hits per ID is 10.34 which is above the limit of acceptable anonymity” ��This is a bit of an easy argumentation. In course of manuscript it is stressed multiple times, how important anonymity is and here the author is implying, that it is ok to not reach absolute anonymity. 

[RESPONSE]: I agree that this presentation was unfortunate. I have reordered the presentation of the results so that the mean (which is ok) is presented first, followed by the min (which is not ok) – leading to a clearer indication that overall, this configuration cannot ensure anonymity for all participants.

REVIEWER #1 (pdf): 32. Further, after reading the Evaluation section, for me there seems to be an inherent conflict between ‘encoding success’ and ‘anonymity’. This is greatly impacting usability, as a user I shouldn’t have to know/understand the details of the CANDIDATE algorithm in order to be able to choose the just correct coding space. 

[RESPONSE]: Yes, it is indeed a trade-off between encoding success and anonymity, and the results shows that a suitable compromise if found with a coding space 10 times the number of participants. This ratio is also now fixed to 10 in the revised tool. The anonymity is also related to the total population of the group (country, region, institution, etc). A warning message with a simple anonymity estimate was added to the tool to make the user more aware of the anonymity for a given study. A comment on this was also added at the end of the section outlining the CANDIDATE algorithm. 

REVIEWER #1 (pdf): 33. The biggest weakness IMHO is not mentioned: The phonebook attacking scenario implies that the attacker got hold of the anonymous data and tries to identify the individuals. However, if an attacker was able to obtain the research data why shouldn’t s/he also be able to get hold of the study organization data where the participant’s name are listed? So the whole method relies on the security of the names list (i.e. the input data). The link-table approach, of course, harbors the same problem of keeping the link-table securer. However, even if an attacker should be able to obtain the participants name list and the anonymous data s/he could not link them. 

[RESPONSE]: The reviewer is indeed right and an important point. If an attacker somehow can be certain that an individual was part of the experiment (from other means) then the attacker can also find the ID of the participant. A paragraph was added to the Discussion section to explicitly elaborate on this point with some suggestions on how to manage this in practice.

REVIEWER #1 (pdf): 34. A further limitation is that the whole evaluation was only conducted with names in Latin letters, whilst in Lines 166-175 it is explicitly stated that any kind of input string could be used. Unfortunately, therefore one can only consider the CANDIDATE tool validated for names provided in Latin-alphabet, of course considering the here found boundaries/limitations. 

[RESPONSE]: Yes, this is a very correct observation indeed. Although the names in the test suite were from all over the world (including Chinese and Arabic names), they were transcribed using Latin characters (as author names often are in international publications). Soundex only works with the Latin alphabet, and Soundex should not be used with other representations such as phone numbers. The text has therefore been revised to reflect this limitation (the Introduction to CANDIDATE pointed out here), the description of Soundex, as well as in the Discussion. The tool has also been updated to clearly indicate the character-coding limitation. 

REVIEWER #1 (pdf): 35. A further practical problem, though not a limitation of CANDIDATE itself, is, that the experimenter must pay great attention for possible errorness Input strings when generating the IDs. 

[RESPONSE]: I fully agree with the reviewer. There is always a risk that input errors caused by the user may lead to problems. The algorithm performs two steps to reduce the chance of error in the input strings. The Soundex algorithm ensures that the tool can handle several types of spelling mistakes as it performs a type of approximate string matching. Second, the name parts (first, middle, second) are sorted in alphabetical order so that it does not matter which order the names are input. Both mechanisms are already briefly explained in the manuscript. A cautionary note about the need for careful input was added to the last sentence of the Discussions section of the revised manuscript.

REVIEWER #1 (pdf): 36. “Evaluations show that the tool successfully assigns unique and anonymous IDs to participants” ��Being picky, this is not true, as assigning unique and anonymous IDs depends on the correct set of parameters (as you write in following sentence). 

[RESPONSE]: Good point. The sentence was moderated with “the tool with appropriate parameters can successfully assign unique and anonymous IDs to participants”.

Reviewer #2 comments:

REVIEWER #2: Reviewer #2: The manuscript by Sandes describes the CANDIDATE tool, a computational method for anonymizing participant identities in research studies. Due to real and perceived privacy concerns, as well as regulations such as the GDPR, there remains a need for simple, robust tools for maintaining the privacy of individuals in research studies, while enabling researchers to potentially link information from the same individual across longitudinal studies or multiple studies.

[RESPONSE]: Thank you. This is a very accurate summary of the manuscript. 

REVIEWER #2: Although I work in a different field and thus I am not familiar with most of the existing tools, I believe the manuscript did a good job of explaining the challenges and existing tools in the field. In particular, the computational modeling of different study sizes and scenarios is necessary to understand the interplay between study size, anonymity, collision rate (same ID assigned to multiple individuals) vulnerability to “phonebook attacks” to decipher identities, and the ID space (length of ID code.) The main strength of the manuscript is the clear description of the tool, and the easy-to-understand examples. In particular, I appreciated the recommendation that the ID pool be approximately 10x the size of the participant list (e.g., encoding 100 participants with a 3-digit number.) A weakness of the manuscript is that the body or discussion could have included some information about extensions of the tool (for example, whether a tool like this could be adapted to work with numerical, or genetic, identity information instead of names.) Also depending on space, some of the tables as have been easier to interpret as graphs of curves. Overall the CANDIDATE tool appears simple to use and useful, and I recommend publication of the manuscript.

[RESPONSE]: Thank you very much for these encouraging comments. A description of the opportunities and limitations of using the tool with generic identity information was added to the discussion (see response to specific comment below). The results tables were replaced with charts for simplified interpretation.

Specific issues:

REVIEWER #2: I think the manuscript could benefit from putting the work into a broader context. A graph or two in place of some tables would make it easier to quickly interpret the study.

[RESPONSE]: The discussion was extended to place the work in a broader context in terms of possibility to use genetic codes/biometric information instead of names. The results tables were replaced with charts for simplified interpretation.

REVIEWER #2: a. The discussion of the need for privacy is excellent, however, the current study is limited to encoding of individuals with roman alphabet names encoded by 26 characters. It is possible that a deeper discussion of the Soundex algorithm could rectify this, however, it is unclear whether the Soundex algorithm can deal with Chinese or Russian names, or whether these would need to be romanized. Alternatively, are there other tools that can sanitize name information or make them more robust, that could be used in place of Soundex tool?

[RESPONSE]: This point was also raised by the other reviewer. The text has been revised to clarify this point (in the section describing the CANDIDATE procedure). In short, the names need to be Romanised to use Soundex (which was designed for English). In principle CANDIDATE may be used with other scripts but then without the error tolerance, or language specific error tolerance mechanisms must be tailor made for the language. The current implementation does however not support non-Latin characters. The tool was also updated with information about this.

REVIEWER #2: b. It would be interesting to learn if the tool could be enhanced or even used with participant IDs that were coded with a combination of numbers and letters, or a combination of names and genetic or biometric information, for example. Alternatively, can or should the CANDIDATE tool be adapted to use an alphanumeric ID space, instead of a strictly numeric ID space as in the examples. While this may be speculative and beyond the scope of the study, there is a clear future need for robust identification of individuals, whilst preserving their privacy.

[RESPONSE]: This is a very interesting and relevant question. In short, if the representation is consistent on different occasions (no variations) it will work with CANDIDATE. If the information varies such as certain biometric information (e.g., fingerprint matching), it will not be possible to use CANDIDATE unless some step is used to reduce or “quantize” the information to a consistent representation. A paragraph addressing this point was added to the end of the Discussion section.

REVIEWER #2: c. The modeling appeared robust for groups of 10-1000 individuals, but it is unclear whether this tool could be used on a larger scales, such as encoding information for 10’s of millions of individuals, or a large nation such as the UK. If this is not possible, it could be useful to provide context of whether CANDIDATE could be useful in subsets of those large biobank studies.

[RESPONSE]: This is a highly relevant question. The anonymity of CANDIDATE comes from the ambiguities that arise from the participant being a subset of a larger population. If one codes an entire population (such as a large biobank dataset with 10’s of millions of individuals) the anonymity of the participants will be reduced. However, CANDIDATE is appropriate for studies involving a subset of such large lists as the coding space will be specific to the list and lead to false positives for other individuals in the superset. A paragraph has been added in the Discussion section to address this point. 

REVIEWER #2: d. I am not familiar with the details of different hash functions. The manuscript did a good job of explaining the goal of the hash, to provide one-way encoding, but it would be nice to have an explanation of different hash functions, and whether most are equivalent, and what are the tradeoffs of using multiple hashes VS single hashes.

[RESPONSE]: It should not be necessary for the reader to be familiar with the details of the hashing algorithms. However, references to sources detailing each of the two basic hash algorithms (djb2 and CRC-32) were added as well as a brief explanation about how the two algorithms work. An example was also added in the revised manuscript (see the new Table 1) to illustrate how the different variations of the hash algorithms modify an input string at the different phases.

REVIEWER #2: Finally, while the language is clear, the manuscript could benefit from copyediting for language and spelling.

[RESPONSE]: The revised manuscript has been carefully reviewed for language issues.

---

## [Decision Letter · Decision Letter 1]

3 Sep 2021

PONE-D-21-20957R1CANDIDATE: A tool for generating anonymous participant-linking IDs in multi-session studiesPLOS ONE

Dear Dr. Sandnes,

Thank you for submitting your manuscript to PLOS ONE. After careful consideration, we feel that it has merit but does not fully meet PLOS ONE’s publication criteria as it currently stands. Therefore, we invite you to submit a revised version of the manuscript that addresses the points raised during the review process.

We look forward to receiving your revised manuscript.

Kind regards,

Anandakumar Haldorai, PhD

Academic Editor

PLOS ONE

Journal Requirements:

Additional Editor Comments:

Recommended for minor revision.

Reviewers' comments:

Reviewer's Responses to Questions

**Comments to the Author**

1. If the authors have adequately addressed your comments raised in a previous round of review and you feel that this manuscript is now acceptable for publication, you may indicate that here to bypass the “Comments to the Author” section, enter your conflict of interest statement in the “Confidential to Editor” section, and submit your "Accept" recommendation.

Reviewer #1: All comments have been addressed

Reviewer #2: All comments have been addressed

2. Is the manuscript technically sound, and do the data support the conclusions?

Reviewer #1: Yes

Reviewer #2: Yes

3. Has the statistical analysis been performed appropriately and rigorously? 

Reviewer #1: Yes

Reviewer #2: Yes

4. Have the authors made all data underlying the findings in their manuscript fully available?

Reviewer #1: Yes

Reviewer #2: Yes

5. Is the manuscript presented in an intelligible fashion and written in standard English?

Reviewer #1: Yes

Reviewer #2: Yes

6. Review Comments to the Author

Reviewer #1: Se attached File.

Reviewer #2: The manuscript is greatly improved. The graphs help make the data more accessible, and the later sections of the paper do a really good job of explaining the limitations and appropriate use-cases for the tool.

7. PLOS authors have the option to publish the peer review history of their article (what does this mean?). If published, this will include your full peer review and any attached files.

Reviewer #1: **Yes: **Mario Lorenz

Reviewer #2: No

---

## [Author Response · Author response to Decision Letter 1]

7 Oct 2021

Response to reviewers’ comments on revision 2

Reviewer #1

[Reviewer #1]: 

1. Line 241: ≥ 9: digest := djb2(name + salt[8 - hash-type]) � Isn’t this bound to run in a run time error as hash-type will be at least 9 resulting in a negative index? However even if inverted “hash-type – 8” It could run into a runtime error as the value of hash-type is not constrained but the salt array has a pre-defined length. I know I am being picky here but as this is basically the core function of your whole approach the possibilities of runtime errors should be completely omitted. Therefore, a predefined salt-array with a fixed set of entries cannot be the solution. I would suggest to generate the salt generically during runtime.

[RESPONSE]: The reviewer is right that there is a risk of a run-time error. The simulations shows that this is highly unlikely (see Table 2). With N = 1000 the hash-type value is never larger than 5, and with larger N’s it is lower. To generate a salt generically during run-time as suggested by the reviewer may also not be a feasible solution. From a theoretical perspective one may not be able to deterministically find a salt that results in a free slot that at the same time provides sufficient obfuscation. In any case the algorithm will detect if such a situation occurs and report to the experimenter that it was unable to encode a given participant. To make this situation more transparent to the reader I have added this as an exception in the pseudo-code, and also added a comment regarding this in the text (just before Table 2).

2. REVIEWER #1 (pdf): 14. “199 FOR EACH (hash-type, validation-code) ATTACHED TO IDoriginal” parameters in FOR EACH not previously defined

[RESPONSE]: The hash-type and validation-code in the FOR EACH are declared in the lines above (in the same IF block)

As far as I see it the If statement where hash-type and validation-code are defined belong to the Add(…) function. The FOR EACH is part of the Lookup(…) so the variables defined in Add(…) should be unknown in Lookup(…).

[RESPONSE]: Actually, the variables defined above were intended as global/state variables to the algorithm. I have adjusted the pseudo code with labels so that it is clearer which are internal state variables to the algorithm, constants, and what are variable parameters.

3. REVIEWER #1 (pdf): 16. I understand why the Sanitize method removes these characters, however maybe an Experimenter relies on these removed characters in order to create unique input strings. In the input text field I would check for the removed characters and make them invalid input.

[RESPONSE]: This is an excellent suggestion. An input check was added to the tool that now only allows variations on the Latin alphabet, numbers (in case of phone numbers), @ and dot (in case of e-mails), and hyphen (as used in some connected names). The manuscript was also updated to reflect this change. 

Unfortunately I was unable to detect this change is the manuscript. Could you please give me direction?

[RESPONSE]: This is perhaps a misunderstanding on my part. The change was done to the browser implementation (updated on GitHub), not the manuscript. In the currently revised manuscriptI have removed sanitize from the algorithm and written that the algorithm assumes sanitized names, and that sanitized names can be ensured using input checks in the user interface.

4. When looking at the hash function I was wondering if there is a reason why crc32 was not used for the letter shifted input names but only djb2?

[RESPONSE]: No, there was no reason for this. Testing showed that there were enough hash functions as is. If one needs more hash functions, all the pre-processing cases that are applied to djb2 (shifts and adding the salt) can also be applied to CRC32 with the desired effect. I have added a comment about this in the text (just before Table 2). 

5. I appreciate the added Fig.1 and Fig2. However, I think Fig2 is currently not really self-containing and is not really understandable. I think it would be good to also use concrete example as in Fig1 for explaining it.

[RESPONSE]: I have replaced Fig 2 with a new illustration (Fig. 3) that shows the steps involved in the encoding (Fig. 3 a, b, c, d, e, f, g and h). This figure builds on the example in Table 3 which now is obsolete. I have therefore removed Table 3 and 4.

6. REVIEWER #1 (pdf): 22. General Comment: Pls add flow charts or sufficient UML diagrams for all algorithms described in section “The CANDIDATE anonymisation tool“. It would allow for drastically shorten text, less repetition and far easier understanding.

[RESPONSE]: A diagram illustrating the essence of the algorithm, namely the collision handling using multiple hash functions was added to the revised manuscript (as this is the core of the approach). A diagram illustrating the generation of the different hash functions was also added. An attempt was made to make it understandable for a wider readership as knowledge and experience is required to read and interpret UML diagrams.

Here I disagree. One who can read AND understand the given pseudocode is highly likely able to comprehend flow charts or sufficient UML diagrams. Further, it will be a big help for colleagues who would like to build on your work. It took me properly 5-time to really think through the algorithm with just the pseudocode and the textual description then it would have been with proper diagrams. By the same time I am also less confident to not have missed a glitch or an error.

[RESPONSE]: I have added flow charts (see new Fig. 1 (a, b c and d). 

Reviewer #2

[Reviewer #2]: The manuscript is greatly improved. The graphs help make the data more accessible, and the later sections of the paper do a really good job of explaining the limitations and appropriate use-cases for the tool.

[RESPONSE]: Thank you very much for these encouraging comments.

General changes

I have added a paragraph in the introduction where the term “anonymous” is briefly discussed and reflected upon which I think is relevant for this study.

---

## [Decision Letter · Decision Letter 2]

13 Oct 2021

PONE-D-21-20957R2CANDIDATE: A tool for generating anonymous participant-linking IDs in multi-session studiesPLOS ONE

Dear Dr. Sandnes,

Thank you for submitting your manuscript to PLOS ONE. After careful consideration, we feel that it has merit but does not fully meet PLOS ONE’s publication criteria as it currently stands. Therefore, we invite you to submit a revised version of the manuscript that addresses the points raised during the review process. Please submit your revised manuscript by Nov 27 2021 11:59PM. If you will need more time than this to complete your revisions, please reply to this message or contact the journal office at plosone@plos.org. Please include the following items when submitting your revised manuscript:A rebuttal letter that responds to each point raised by the academic editor and reviewer(s). You should upload this letter as a separate file labeled 'Response to Reviewers'.A marked-up copy of your manuscript that highlights changes made to the original version. You should upload this as a separate file labeled 'Revised Manuscript with Track Changes'.An unmarked version of your revised paper without tracked changes. You should upload this as a separate file labeled 'Manuscript'.If applicable, we recommend that you deposit your laboratory protocols in protocols.io to enhance the reproducibility of your results. Protocols.io assigns your protocol its own identifier (DOI) so that it can be cited independently in the future. For instructions see: https://journals.plos.org/plosone/s/submission-guidelines#loc-laboratory-protocols. Additionally, PLOS ONE offers an option for publishing peer-reviewed Lab Protocol articles, which describe protocols hosted on protocols.io. Read more information on sharing protocols at https://plos.org/protocols?utm_medium=editorial-email&utm_source=authorletters&utm_campaign=protocols.

We look forward to receiving your revised manuscript.

Kind regards,

Anandakumar Haldorai, PhD

Academic Editor

PLOS ONE

Journal Requirements:

Additional Editor Comments (if provided):

The figures are needs to improve. The quality of figures maybe improved with proper image editor.

Reviewers' comments:

Reviewer's Responses to Questions

**Comments to the Author**

1. If the authors have adequately addressed your comments raised in a previous round of review and you feel that this manuscript is now acceptable for publication, you may indicate that here to bypass the “Comments to the Author” section, enter your conflict of interest statement in the “Confidential to Editor” section, and submit your "Accept" recommendation.

Reviewer #1: All comments have been addressed

Reviewer #2: All comments have been addressed

2. Is the manuscript technically sound, and do the data support the conclusions?

Reviewer #1: Yes

Reviewer #2: Yes

3. Has the statistical analysis been performed appropriately and rigorously? 

Reviewer #1: Yes

Reviewer #2: Yes

4. Have the authors made all data underlying the findings in their manuscript fully available?

Reviewer #1: Yes

Reviewer #2: Yes

5. Is the manuscript presented in an intelligible fashion and written in standard English?

Reviewer #1: Yes

Reviewer #2: Yes

6. Review Comments to the Author

Reviewer #1: Thank you very much for the revision. Just one minor Editing issues: Fig 1a and 1c need minor polishing so that the text is not intersected by lines and all arrows are straight.

Reviewer #2: The revised manuscript is relatively easy to follow, the explicit examples in Figures 1 and 3 are good and the added discussion of anonymity is helpful. I have only two minor comments. One cosmetic issue is that the flow chart would look better and be easier to follow if the text inside the different flow diagrams in Fig 1a and 1c was resized to make it more readable. Second, for the the online version of the algorithm, it would be helpful to add a hyperlink to the github page (especially if the code is expanded or modified in the future.

7. PLOS authors have the option to publish the peer review history of their article (what does this mean?). If published, this will include your full peer review and any attached files.

Reviewer #1: **Yes: **Mario Lorenz

Reviewer #2: No

---

## [Author Response · Author response to Decision Letter 2]

28 Oct 2021

Response memo

EDIROR: The figures are needs to improve. The quality of figures maybe improved with proper image editor.

RESPONSE: The issues raised by the reviewers have been fixed (see below).

Reviewer #1: Thank you very much for the revision. Just one minor Editing issues: Fig 1a and 1c need minor polishing so that the text is not intersected by lines and all arrows are straight.

RESPONSE: Fig 1a and 1c have been adjusted to avoid collisions between text and lines. All arrows have been straightened.

Reviewer #2: The revised manuscript is relatively easy to follow, the explicit examples in Figures 1 and 3 are good and the added discussion of anonymity is helpful. I have only two minor comments. One cosmetic issue is that the flow chart would look better and be easier to follow if the text inside the different flow diagrams in Fig 1a and 1c was resized to make it more readable. Second, for the the online version of the algorithm, it would be helpful to add a hyperlink to the github page (especially if the code is expanded or modified in the future.

RESPONSE: The problems in Fig 1a and 1c were fixed by making the boxes larger so ensure a consistent text size.

A link to the github page is added in the algorithm caption.

---

## [Decision Letter · Decision Letter 3]

15 Nov 2021

CANDIDATE: A tool for generating anonymous participant-linking IDs in multi-session studies

PONE-D-21-20957R3

Dear Dr. Sandnes,

We’re pleased to inform you that your manuscript has been judged scientifically suitable for publication and will be formally accepted for publication once it meets all outstanding technical requirements.

Kind regards,

Anandakumar Haldorai, PhD

Academic Editor

PLOS ONE

Additional Editor Comments (optional):

Recommended.

Reviewers' comments:

Reviewer's Responses to Questions

**Comments to the Author**

1. If the authors have adequately addressed your comments raised in a previous round of review and you feel that this manuscript is now acceptable for publication, you may indicate that here to bypass the “Comments to the Author” section, enter your conflict of interest statement in the “Confidential to Editor” section, and submit your "Accept" recommendation.

Reviewer #1: All comments have been addressed

Reviewer #2: All comments have been addressed

2. Is the manuscript technically sound, and do the data support the conclusions?

Reviewer #1: Yes

Reviewer #2: Yes

3. Has the statistical analysis been performed appropriately and rigorously? 

Reviewer #1: Yes

Reviewer #2: Yes

4. Have the authors made all data underlying the findings in their manuscript fully available?

Reviewer #1: Yes

Reviewer #2: Yes

5. Is the manuscript presented in an intelligible fashion and written in standard English?

Reviewer #1: Yes

Reviewer #2: Yes

6. Review Comments to the Author

Reviewer #1: (No Response)

Reviewer #2: (No Response)

7. PLOS authors have the option to publish the peer review history of their article (what does this mean?). If published, this will include your full peer review and any attached files.

Reviewer #1: **Yes: **Mario Lorenz

Reviewer #2: No

---

## [Editor Report · Acceptance letter]

2 Dec 2021

PONE-D-21-20957R3 

CANDIDATE: A tool for generating anonymous participant-linking IDs in multi-session studies 

Dear Dr. Sandnes:

I'm pleased to inform you that your manuscript has been deemed suitable for publication in PLOS ONE. Congratulations! Your manuscript is now with our production department. 

Kind regards, 

on behalf of

Dr. Anandakumar Haldorai 

Academic Editor

PLOS ONE